

# The 8.2 ka event in northern Spain: timing, structure and climatic impact from a multi-proxy speleothem record

Hege Kilhavn[1,2], Isabelle Couchoud[1,2], Russell N. Drysdale[2], Carlos Rossi[3], John Hellstrom[2], Fabien Arnaud[1], Henri Wong[4]

[1]EDYTEM, Université Savoie Mont Blanc, 73376 Le Bourget du Lac, France
[2]School of Geography, Earth and Atmospheric Sciences, The University of Melbourne, 3010 Victoria, Australia
[3]Dept. Petrología y Geoquímica, Facultad de Ciencias Geológicas, Universidad Complutense, 28040 Madrid, Spain
[4]Australian Nuclear Science and Technology Organisation, Lucas Heights, NSW 2234, Australia

*Correspondence to*: Hege Kilhavn (hege.kilhavn@univ-smb.fr)

**Abstract.** The 8.2 ka event is regarded as the most prominent climate anomaly of the Holocene, and is thought to have been triggered by a meltwater release to the North Atlantic that was of sufficient magnitude to disrupt the Atlantic Meridional Overturning Circulation (AMOC). It is most clearly captured in Greenland ice-core records, where it is reported as a cold and dry anomaly lasting ~160 years, from $8.25 \pm 0.05$ ka BP until $8.09 \pm 0.05$ ka BP (Thomas et al., 2007). It is also recorded in several archives in the North Atlantic region, however its interpreted timing, evolution and impacts vary significantly. This inconsistency is commonly attributed to poorly constrained chronologies and/or inadequately resolved time series. Here we present a high-resolution speleothem record of early Holocene palaeoclimate from El Soplao Cave in northern Spain, a region pertinent to studying the impacts of AMOC perturbations on south-western Europe. We explore the timing and impact of the 8.2 ka event on a decadal scale by coupling speleothem stable carbon and oxygen isotopic ratios, trace element ratios (Mg/Ca and Sr/Ca) and growth rate. Throughout the entire speleothem record, $\delta^{18}$O variability is related to changes in effective recharge. This is supported by the pattern of changes in $\delta^{13}$C, Mg/Ca and growth rate. The 8.2 ka event is marked as a centennial-scale negative excursion in El Soplao $\delta^{18}$O, starting at $8.19 \pm 0.06$ ka BP and lasting until $8.05 \pm 0.05$ ka BP, suggesting increased recharge at the time. Although this is supported by the other proxies, the amplitude of the changes is minor and largely within the realm of variability over the preceding 1000 years. Further, the shift to lower $\delta^{18}$O leads the other proxies, which we interpret as the imprint of the change in the isotopic composition of the moisture source, associated with the meltwater flux to the North Atlantic. A comparison with other well-dated records from south-western Europe reveals that the timing of the 8.2 ka event was synchronous, with an error-weighted mean age for the onset of $8.23 \pm 0.03$ ka BP and $8.10 \pm 0.05$ ka BP for the end of the event. This compares favourably with the NGRIP record. The comparison also reveals that the El Soplao $\delta^{18}$O is structurally similar to the other archives in south-western Europe, and the NGRIP ice-core record.



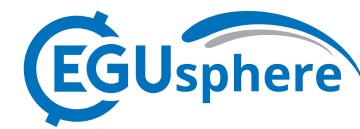

## 1 Introduction

Compared to the Last Glacial period, with its large swings between stadial and interstadial states, the climate of the Holocene has been relatively stable (Denton and Karlén, 1973; Bond et al., 2001; Mayewski et al., 2004). Notwithstanding this, some abrupt climatic changes have occurred, notably the 8.2 ka (Alley et al., 1997) and 4.2 ka events (Weiss et al., 1993; Drysdale et al., 2006). The 8.2 ka event was short-lived (~160-year) and represents the most prominent climate anomaly of the Holocene (Alley et al., 1997; Alley and Ágústsdóttir, 2005; Rohling and Pälike, 2005; Thomas et al., 2007), at least for the Northern

Hemisphere. A widely accepted explanation for the event is a large outburst of freshwater from proglacial lakes Agassiz and Ojibway during the final retreat phase of the Laurentide ice sheet (Barber et al., 1999; Li et al., 2012; Törnqvist and Hijma, 2012). An alternative cause, a freshwater outburst from the collapsing ice saddle over Hudson Bay, has also been proposed (Gregoire et al., 2012; Matero et al., 2017). Additionally, there is evidence for multiple meltwater fluxes prior to and/or during the event (e.g. Ellison et al., 2006; Hillaire-Marcel et al., 2007; Lochte et al., 2019; Brouard et al., 2021). Whilst the exact

trigger is still debated, the volume and routing of freshwater entering the North Atlantic Ocean was apparently sufficient to disrupt the Atlantic Meridional Overturning Circulation (AMOC), which transports heat from the tropics to the Arctic region (Barber et al., 1999; Ellison et al., 2006).

  The 8.2 ka event is most clearly captured by Greenland ice cores (Alley et al., 1997; Thomas et al., 2007). According to the most recent ice-core age model (GICC05) (Rasmussen et al., 2006), and based on oxygen isotope ratios in the ice ($\delta^{18}O_{ice}$),

event started at 8.25 ka BP and lasted until 8.09 ka BP (where BP is before present, and present is AD 1950), with a counting uncertainty of $\pm$ 47 years (North Grip Ice Core Project Members, 2004; Thomas et al., 2007). In Greenland ice cores, the event is asymmetric in shape, and preserves evidence of significant decadal variability. Its onset in the NGRIP record is defined by a sharp decrease in $\delta^{18}O_{ice}$ (of more than 1 ‰), followed by an interval of fluctuating values that are consistently below the mean of the preceding 1000 years. Within this interval are three abrupt increases at 8.23, 8.18 and 8.14 ka BP, defining the

"central event" (a double-trough) of lowest $\delta^{18}O_{ice}$ values, lasting from 8.23 to 8.14 ka BP. This is followed by a gradual recovery to pre-event $\delta^{18}O_{ice}$ values (North Grip Ice Core Project Members, 2004; Thomas et al., 2007).

  Climate over Greenland throughout the 8.2 ka event became colder (inferred from decreased $\delta^{18}O_{ice}$) and drier (inferred from a decrease in snow accumulation rate). The concomitant increase in wind-blown dust, sea salt and forest-fire soot, and decrease in methane, in the ice cores suggest that the event was widespread (Alley et al., 1997; Alley and Ágústsdóttir, 2005). However,

event timing, duration, shape and impacts, as reconstructed from various palaeoclimate archives across Europe, vary significantly. This inconsistency is commonly attributed to poorly constrained chronologies and/or low-resolution time series. Further, the 8.2 ka event is thought to be superimposed on a longer-term anomaly related to solar variability (Bond et al., 2001; Rohling and Pälike, 2005), which complicates the correlation between high- and low-resolution archives. Nevertheless, the 8.2 ka event *sensu stricto* as "related to a freshwater outburst in the North Atlantic" has been recorded in numerous archives

across Europe (Alley and Ágústsdóttir, 2005; Morill et al., 2013).



One of the main challenges for reconstructing palaeoclimate during the 8.2 ka event is the lack of precise and accurate high-resolution records that can be correlated with Greenland ice cores. In marine sediments, chronologies can be hampered by the uncertainties in the radiocarbon reservoir correction. The marine reservoir effect varies spatially and temporally, and can be corrected using marine radiocarbon age calibration curves (e.g. Marine20), with the most recent one having uncertainties less

than 200 [14]C years through the Holocene (Heaton et al., 2020). In lake sediments, a lack of datable material can be a problem, although this is circumvented where varves are present, which can be used to construct chronologies of high precision and accuracy (e.g. Brauer et al., 1999; von Grafenstein et al., 1999; Andersen et al., 2017). With speleothems, it is possible to construct precise and accurate chronologies with U-series dating; where high-resolution proxy series are available, these archives can be excellent for correlating abrupt climate changes across different climatic regions. Speleothem $\delta^{18}$O is often

used as the climate proxy that connects speleothems to other archives, such as ice cores and deep-sea sediments. However, disentangling the controlling factors of speleothem $\delta^{18}$O is complex, and a combination of several speleothem properties (e.g. stable isotopes, trace elements, growth rate) are often required to overcome or minimise the uncertainties of $\delta^{18}$O interpretation. In south-western Europe, numerous archives record climate variability through the Holocene, some of which preserve evidence for a climate anomaly around 8.2 ka. Due to its proximity to the North Atlantic, south-western Europe is a climatologically

important region for studying the impact of AMOC perturbations (Baldini et al., 2015). Several high-quality speleothem records describe the Holocene climate of this region in terms of rainfall amount and/or seasonality (Domínguez-Villar et al., 2008; Railsback et al., 2011; Smith et al., 2016; Moreno et al., 2017; Baldini et al., 2019; Benson et al., 2021). Specifically, the regional climatic impact of the 8.2 ka event has mainly been described in terms of hydrological change from these studies. The majority of regional studies reports the event as a wet interval (e.g. Railsback et al., 2011; Baldini et al., 2019; Benson et

al., 2021), although with conflicting interpretations of whether it was a change in total annual rainfall amount or changes in the seasonal distribution of rainfall (i.e. more winter rainfall), whilst some ambiguity remains over event timing. Additionally, in Kaite Cave (Northern Spain) two $\delta^{18}$O anomalies, separated by ~140 years, have been interpreted to trace modifications of the isotopic composition of the ocean surface in response to successive discharges of isotopically depleted meltwaters of lakes Agassiz and Ojibway (Domínguez-Villar et al., 2008). In other regional archives, such as lake records, a longer-term change

in hydrology around 8.2 ka is usually reported, interpreted as a shift to wetter conditions, potentially related to summer insolation rather than to the 8.2 ka event (i.e. the outburst of freshwater) itself (Morellón et al., 2018). Taken together, it appears that detailed palaeoclimate data on spatial or seasonal impacts and connections between the North Atlantic (e.g. perturbations of the AMOC) and continental climate (e.g. precipitation) at the 8.2 ka event in south-western Europe are not yet clearly understood.

To address this, we present a high-resolution, multi-proxy study of a stalagmite from El Soplao Cave in northern Spain. We explore the hydrological impacts (including the role of seasonality) of the 8.2 ka event at a decadal scale by coupling stable carbon and oxygen isotopic ratios, trace element ratios (Mg/Ca and Sr/Ca) and growth rate. Additionally, we compile and standardise high-quality archives from south-western Europe and northern Morocco (fig. 1) to examine the consistency of timing, duration, shape and impact of the 8.2 ka event.






**Figure 1:** Location of the main palaeoclimate records discussed in the text: Soplao (El Soplao Cave: this study); Garma (La Garma Cave: Baldini et al., 2019); Kaite (Kaite Cave: Domínguez-Villar et al., 2009; 2017); GdL (Galeria das Lâminas: Benson et al., 2021); Chaara (Chaara Cave: Ait Brahim et al., 2019); Asiul (Cueva de Asiul: Smith et al., 2016); Arcoia (Cova da Arcoia: Railsback et al., 2011); Villars (Villars Cave: Ruan, 2016). Base map (ETOPO1 1 Arc-Minute Global Relief Model) collected from NOAA National

Geophysical Data Center and map created in QGIS.

## 2 Cave setting

El Soplao Cave is located approximately 12 km from the coastline in the Arnero Sierra of northern Spain (fig. 1). The cave is situated in the northern watershed of the wet-temperate Cantabrian mountains. Most of the region's rainfall originates from the North Atlantic (Ancell and Célis, 2013; Rossi et al., 2018), with an average annual amount of 1271 ± 169 mm; mean annual

air temperature is 12.7 ºC (Ancell and Célis, 2013) (data for 1981-2010 CE). At the entrance of El Soplao Cave (550 m a.s.l.), the mean annual air temperature is 11.3 ºC (Rossi et al., 2018). Meteorological data from the nearest GNIP station, Santander (52 m a.s.l., ~65 km from El Soplao Cave), indicate a mean annual air temperature of 14.8 ºC and a mean annual precipitation



of 1050 ± 140 mm (collection period: 2000-2015 AD; Rodríguez-Arévalo et al., 2011). Comparison between the Cantabrian province and Santander GNIP station data reveals similar seasonal patterns, with the rainiest part of the year occurring between

autumn and spring, with a maximum in November (suppl. fig. S1). The Santander GNIP station also provides monthly $\delta^{18}O$ values of precipitation, which are positively correlated with temperature and negatively correlated with precipitation. However, due to the predominance of winter rainfall in the region, either or both of the temperature and rainfall amount effects may be spurious. Deconvolution of monthly rainfall $\delta^{18}O$, rainfall amount and mean temperature data suggest that neither of the effects have a persistent influence on rainfall $\delta^{18}O$ but are rather transient features (suppl. fig. S2).

El Soplao Cave consists of 23 km of surveyed passages formed in dolostones that are sandwiched between limestones. A detailed description of the cave is given by Rossi and Lozano (2016) and Rossi et al. (2018). The stalagmite studied here, SIR-14, was collected from the SIR passage of the cave system located ~120 m below the surface and ~1 km from both of the natural cave entrances (fig. 2). To minimize the geoheritage impact on the cave, an exact replica was emplaced in the same position in which SIR-14 grew (Baeza et al., 2018: SIR-14 is coded as "Replica Soplao-1"). The SIR-14 stalagmite is ~700

mm long and was apparently inactive when collected and during the monitoring period. However, it should be noted that the drip above the stalagmite has recently been reactivated. The stalagmite grew from 73.5 ka BP until 6.3 ka BP, but the focus of this study is the Holocene portion (the upper ~400 mm). During its growth, the stalagmite was fed by a stalactite whose morphology is a mixture of bulbous and thin soda straw (fig. 2). Stalagmite SIR-1 (Rossi et al., 2018) was located in the same passage and grew ~5 m away from SIR-14 (fig. 2). Stalagmite SIR-1 is located at a slightly lower elevation in a slope of a

vadose trench which currently has an ephemeral stream at the bottom. This relative difference in elevation makes SIR-14 more protected against flooding, which can induce deposition of clay on the stalagmite surface and cause dissolution. The catchment area of the dripwaters in the SIR passage mainly consists of rugged dolostone outcrops with shrubland vegetation (Rossi et al., 2018).

Rossi and Lozano (2016) studied the hydrochemistry of ~50 stalagmite-precipitating drips across the cave system over a 3-

year monitoring period (July 2010 to July 2013). In the SIR passage, the relative humidity is close to saturation and air temperature is 10.8 ± 0.2 ºC throughout the year. The passage is located midway between the cave entrances, which drive summer and winter air into the cave; therefore, the cave temperature at the SIR-14 site is not significantly biased towards any season and likely reflects mean annual temperature at the surface above the cave (Rossi and Lozano, 2016). The $CO_2$ of the cave air typically ranges between 450 and 650 ppmv, with minimum values occurring during winter (Rossi and Lozano, 2016).

Dripwaters in the SIR passage have a lower supersaturation with respect to calcite compared to average dripwater values elsewhere in the cave (Rossi and Lozano, 2016). The average Mg/Ca of stalagmite-precipitating dripwaters in the SIR passage is 0.6 ± 0.02 mol mol$^{-1}$. The SIR dripwaters show relatively minor seasonal Mg/Ca change but vary from drip to drip (0.5-0.8 mol mol$^{-1}$), being inversely correlated with drip discharge. This implies that drip rate is a major control on the Mg/Ca ratios of dripwater and consequently on the Mg concentration of the corresponding speleothems (Rossi and Lozano, 2016; Rossi et al.,

2018). The values of $\delta^{18}O$ in dripwater in the cave system range from -7.1 ‰ to -6.3 ‰, with an average in the SIR passage of -6.8 ± 0.02 ‰. The isotopic data plot near the local meteoric water line (LMWL) derived from Santander GNIP station data





(Rossi and Lozano, 2016). The proximity to the LMWL suggests minor soil evaporation, which is consistent with the wet-temperate climate. There is no significant variability between the dripwater samples collected from different seasons: the values reported in Rossi and Lozano (2016) are all within measurement error (0.1 ‰). In the SIR passage, five of the six

monitored drips show relatively constant discharge (coefficient of variation < 17 %), with one drip showing higher variability (57 %). Drip site GASa, located ~100 m from the SIR site, also shows notable variation (coefficient of variation: 31 %) (Rossi and Lozano, 2016). The relatively constant discharge of most SIR drips indicates that a diffuse-flow component is predominant in the vadose zone above the passage. This is likely related to the presence of significant matrix porosity in the dolostone host rock. However, some drips in the SIR passage and nearby sites (GASa) show significant intra-annual/seasonal variations in

discharge, indicating an additional fracture-flow component (Rossi and Lozano, 2016).





**Figure 2: A)** Aerial view of El Soplao Cave system superimposed over the terrain (aerial ortho-image collected from PNOA©Instituto Geográfico Nacional). **B)** Cross-section of the cave system. The yellow star indicates the location of the SIR passage. **C)** Photo of the gallery where stalagmites SIR-14 (this study) and SIR-1 (Rossi et al., 2018) were collected. **D)** Photo of stalagmite SIR-14 *in situ* and the bulbous-shaped feeding stalagmite. Photos: Carlos Rossi.



## 3 Methods

### 3.1 Material

A ~1.5 cm thick slab incorporating the central growth axis was cut from SIR-14 using a diamond saw. The surface was manually grounded and incipiently polished using resin-bonded disks, avoiding abrasive powders that could penetrate into
sample porosity, to reveal internal laminations, macroscopic structures and possible hiatuses, then scanned prior to any sampling. The stalagmite is ~700 mm long, and tapers from the base (~150 mm) to the top (~50 mm). In this study, only the upper ~400 mm is discussed. The laminae are curved with a flattened surface in the axial part. They are thin and faint, and in some places, cannot be distinguished by the naked eye. The stalagmite consists of a combination of translucent-to-whitish calcite and yellowish calcite in the early Holocene section, and a few short sections of darker and less porous calcite. The two
most prominent sections with darker calcite are shown in fig. 3B1 and B2. The stalagmite has several mm-to-cm-scale pores or vugs in the central axis, with the majority being located in the first ~300 mm from the base (i.e. the Pleistocene section). These are likely to represent primary growth features rather than products of dissolution, as shown in stalagmite SIR-1 (Rossi et al., 2018). There are no visible surfaces that could represent hiatuses in the upper section, suggesting it grew continuously through the first half of the Holocene. However, there are some shifts in growth-axis direction, which could be due to some
adjustment of the substratum or the feeding drips (such as a plugged fissure or a shift in the feeding stalactite).

### 3.2 U-Th dating

The chronology of stalagmite SIR-14 was constrained by 21 U-Th dates (suppl. Table S1). Most of the U-Th samples were collected as powders using a 1-mm drill bit fitted to a micromilling instrument, and others (indicated by italics, suppl. Table
S1) were collected using a handheld drill. The samples were drilled from the same slab as the stable isotopes, so that their positions could be coupled directly to the depth profile of the isotope series (fig. 3A). The U-Th samples were prepared for chemical preparation using the procedure of Hellstrom (2003). Following this, the U-Th samples were analysed and their activity ratios determined using a Nu Instruments Plasma multi-collector inductively coupled plasma mass spectrometer (MC-ICP-MS) at the University of Melbourne. Uncorrected ages were calculated using the latest decay constants (Cheng et al.,
2013). Five new dating samples were also measured from stalagmite SIR-1 (Rossi et al., 2018) to improve the chronology in the early Holocene part of this speleothem. Based on these new dates, and the data from the SIR-14 stalagmite, an initial $^{230}Th/^{232}Th$ ratio of 2.7 ± 0.9 was derived for both SIR stalagmites based on modelling. This value is higher than expected for a cool, temperate setting, but is consistently constrained by three samples from SIR-14 with high $^{232}Th/^{238}U$ values using the stratigraphic-constraint approach of Hellstrom (2006). The corrected ages and their standard error (95 % uncertainties) were
used to create a depth-age model (fig. 3C) using the Finite Positive Growth Rate Model, which is based on Monte-Carlo simulations (Corrick et al., 2020). The new U-Th dates for SIR-1 were combined with published ages (Rossi et al., 2018) to





build a revised age model, which was used to update the anchoring points of this speleothem's floating lamina chronology (see suppl. Table S2 and fig. S7 for further details).

### 190   3.3 Stable carbon and oxygen isotopes

Stable oxygen and carbon isotope ratios for stalagmite SIR-14 were obtained by sampling the upper ~400 mm of the stalagmite continuously at 1 mm and perpendicular to the laminae (fig. 3A). The samples were drilled at the University of Melbourne using a Taig CNC micromilling machine fitted with a 1 mm drill bit. The $\delta^{13}$C and $\delta^{18}$O samples were measured on 0.7-0.8 mg aliquots and analysed on an Analytical Precision AP2003 continuous-flow isotope-ratio mass spectrometer at the
University of Melbourne following the procedure of Drysdale et al. (2007). The stable isotope values were normalised to the Vienna Pee Dee Belemnite (VPDB) scale using an internal standard of Carrara marble (NEW1), which was previously calibrated against the international standards NBS18 and NBS19. The mean analytical precision is 0.05 ‰ for $\delta^{13}$C and 0.1 ‰ for $\delta^{18}$O. Samples were also drilled along four growth laminae in order to check if calcite precipitated at or close to isotopic equilibrium (Hendy, 1971).


### 3.4 Trace elements

Residual powders from the stable isotope analyses were prepared for trace element (Mg/Ca and Sr/Ca) measurement at the Australian Nuclear Science and Technology Organisation (ANSTO, Sydney, Australia) using inductively coupled plasma-atomic emission spectroscopy (ICP-AES) (de Villiers et al., 2002). Prior to analysis, the samples were dissolved in a 3 % vol
vol$^{-1}$ (v v$^{-1}$) nitric acid solution to a ratio of 1 mg 5 mL$^{-1}$ to minimize concentration and matrix variation. A standard solution (continuous calibration verification) was run between blocks of five samples to correct for instrumental drift, with a precision of 0.2 % for Mg/Ca and 0.8 % for Sr/Ca. In-laboratory blanks (3 % v v$^{-1}$ HNO$_3$) were also run alongside the samples to check for any contamination contribution to the samples. Additionally, a suite of mixed standards containing a similar Ca concentration to the samples, but with variable amounts of Mg and Sr, were analysed (de Villiers et al., 2002). The final ratios
were manually calculated using the calibration curve of intensity ratio (fig. A1 and A2 in de Villiers et al. (2002)), where the ratios are regressed against the Mg/Ca and Sr/Ca content of the standard solutions (de Villiers et al., 2002).



**Figure 3: A) Polished section of stalagmite SIR-14. Isotopic transects are shown as blue lines; U-Th samples are indicated in white and numbered, corresponding ages (in ka BP) are indicated to the left. Location of the laminae where the Hendy tests were**

**performed are indicated by red lines. B) Sections B1 and B2 show two layers of dense and dark calcite where the growth rate was relatively low (between 9.9-9.1 ka BP and 7.9-6.7 ka BP). C) The age-depth model for stalagmite SIR-14 created using the Finite**



**Positive Growth Rate Model (Corrick et al., 2020). The light and dark red envelopes represent respectively the 66% and 95% confidence intervals. The date with a green symbol was treated as an outlier and excluded from the age modelling.**

## 4 Results

### 4.1 U-Th, age model and growth rate results

In the upper ~400 mm (the Holocene period), all except one of the 21 U-Th age determinations on SIR-14 are in stratigraphic order (within uncertainties). The age profile indicates that this upper portion grew continuously between $11.7 \pm 0.1$ ka BP and $6.4 \pm 0.3$ ka BP. The outlier (at 138.5 mm, 8.5 ka BP) is not linked to any obvious petrographic alterations, nor any anomalously high detrital contamination. It is likely that this sample is older than expected due to localized open-system behaviour leading to U-leaching (Bajo et al., 2016), as inferred for several of the samples in the SIR-1 stalagmite which grew very close to SIR-14 (fig. 2) (Rossi et al., 2018). Uranium concentrations range between 164 and 412 ng g$^{-1}$, averaging 293 ng g$^{-1}$ (suppl. Table S1). The $^{230}$Th/$^{232}$Th activity ratios vary from 23 to 3838, but the majority are high (average of 1142), indicating small-to-negligible influence of detrital thorium. The initial $^{234}$U/$^{238}$U activity ratios vary between 1.47 and 1.35, with the highest values occurring at 11.7 ka BP. There is a sharp decrease in these values towards 11 ka BP, after which the values remain low and stable with only minor variations through the Holocene (suppl. Table S1). The growth rate varies from 12.4 to 224.2 mm ka$^{-1}$, averaging 76.7 mm ka$^{-1}$. The lowest growth rates occur between 9.9-9.1 ka BP and 7.9-6.7 ka BP (less than 50 mm ka$^{-1}$); they correspond to layers of relatively dark and dense calcite (fig. 3B1 and B2). The highest growth rates occur at 9.0 ka BP, 8.3 ka BP, 8.1 ka BP and 6.6 ka BP (> 100 mm ka$^{-1}$); the calcite is more porous and fractured here compared to the sections below and above, and consists of translucent-to-whitish layers (fig. 3A). The growth rate peaks occur just prior to the onset of the 8.2 ka event (at 8.3 ka) and at the very end of the event (at 8.1 ka), with a lower growth rate at the beginning of the event (fig. 4). The growth rate has a moderate, but statistically significant, negative correlation with $\delta^{18}$O, $\delta^{13}$C and Mg/Ca (Table 1).

### 4.2 Variations in Mg/Ca and Sr/Ca

The Mg/Ca values vary from 11.0 to 29.0 mmol mol$^{-1}$ with an average of 15.2 mmol mol$^{-1}$; the Sr/Ca values vary from 0.06 to 0.10 mmol mol$^{-1}$ with an average of 0.08 mmol mol$^{-1}$ (fig. 4). The most outstanding feature in the Mg/Ca profile is the two different ranges of values: higher values occurring during the slow growth periods between 9.9-9.1 ka BP and 7.9-6.7 ka BP (fig. 3B1 and B2), and low values occurring during periods with average to higher growth rates (fig. 4). Through the 8.2 ka interval, the Mg/Ca displays a negative excursion, of low amplitude, and it does not stand out compared to the rest of the record (fig. 4). The main trends of the Mg/Ca profile correlate well with the $\delta^{18}$O and $\delta^{13}$C ($r = 0.74$ and $r = 0.86$, respectively; Table 1).




The Sr/Ca ratio is very low and the signal is variable across the studied period; in particular, it displays two negative excursions coinciding with the prominent peaks in the other proxies, between 9.9-9.1 ka BP and 7.9-6.7 ka BP (fig. 4). Correlation between Sr/Ca and the other measured proxies is generally low ($r < 0.3$; Table 1); however, visual observation indicates a negative

correlation, especially during the slow growth intervals. The Sr/Ca signal is relatively noisy across the 8.2 ka interval with no clear indication of a climate event. This could be due to the relatively low Sr content in the stalagmite, with concentrations being close to the detection limits for the method. Accordingly, Sr/Ca will not be discussed further in the ensuing discussion.

## 4.3 Variations in $\delta^{13}C$ and $\delta^{18}O$

The $\delta^{18}O$ and $\delta^{13}C$ time series yield an average resolution of $13 \pm 10$ years across the Holocene, ranging from 2 to 66 years. The $\delta^{18}O$ values vary between -5.9 and -4.3 ‰ with an average of $-5.2 \pm 0.2$ ‰; the $\delta^{13}C$ values vary between -8.3 and -5.1 ‰ with an average of $-7.3 \pm 0.5$ ‰. The most negative value appears at 8.1 ka BP and is the lowest value in a centennial-scale negative excursion at the time of the 8.2 ka event (fig. 4). The $\delta^{13}C$ profile follows the same general trends as the $\delta^{18}O$ ($r = 0.68$; Table 1), but the values attained at the time of the 8.2 ka event are not the lowest throughout the record, and the minimum

value occur just after aforementioned $\delta^{18}O$ minimum.  The $\delta^{18}O$ and $\delta^{13}C$ signals also show two prominent peaks, between 9.9-9.1 ka BP and 7.9-6.7 ka BP, when the growth rates were at their lowest (fig. 3B1 and B2).

We calculated the 30-point running correlation coefficient for the measured proxies, which highlights time intervals of stronger correlation coefficients. The highest correlation coefficient between $\delta^{18}O$ and $\delta^{13}C$ occurs between 10 and 9 ka and around 8 ka BP ($r > 0.6$), whilst the correlation is relatively low for the rest of the record ($r < 0.6$; (suppl. fig. S3). The highest correlation

coefficients occur in intervals with a large change in the growth rate (fig. 4). Between 10 and 9 ka BP, a period of slow growth, the growth rate has a high negative correlation with $\delta^{18}O$, $\delta^{13}C$ and Mg/Ca and a high positive correlation with Sr/Ca (Table. 1, suppl. fig. S3). The highest negative correlation coefficient between growth rate and $\delta^{18}O$, $\delta^{13}C$ and Mg/Ca ($r > -0.8$) occurs at ~8.0 ka.

## 270    4.4 Isotopic equilibrium

Kinetic effects can influence the isotopic values in speleothems (McDermott, 2004; Lachniet, 2009). Conditions of isotopic equilibrium, which are likely to be met when sufficient time is available for isotope exchange reactions to proceed, can be interrogated by applying the "Hendy test" (Hendy, 1971). This was performed on several laminae in stalagmite SIR-14. The $\delta^{18}O$ and $\delta^{13}C$ display moderate to negligible covariation within the laminae (suppl. fig. S4). There is a scatter of up to 0.6 ‰

in the $\delta^{18}O$ values within a single lamina, suggesting that sampling might have been slightly inaccurate, incorporating powder from different laminae due to the thin and faint lamination in stalagmite SIR-14. However, the enrichment is less than 0.2 ‰ in the $\delta^{18}O$ values from the central axis towards the flanks of the stalagmite, arguing for acceptable near-equilibrium isotopic conditions during precipitation. Additionally, it has been shown that even if the test suggests kinetic fractionation along a




growth lamina, isotope samples located close to the growth axis tend to be closer to equilibrium (McDermott, 2004; Lachniet,

2009). There is a relatively strong covariation ($r = 0.59$, p < 0.001) across the whole record between the $\delta^{18}O$ and $\delta^{13}C$ along the growth axis, presumably related to the fact that both isotope ratios are being shifted in the same direction by common or climate related processes.

Another test for equilibrium conditions during calcite precipitation was performed on modern calcite in El Soplao Cave by comparing the measured fractionation with the theoretical equilibrium fractionation at the cave temperature (Rossi and Lozano,

2016). Results from this test suggest equilibrium or near-equilibrium conditions throughout the cave in the present environment, including the SIR passage (Rossi and Lozano, 2016). This is supported by moderate and stable supersaturation values ($SI_{calcite} = 0.4 \pm 0.1$, n = 6) of the dripwaters feeding stalagmites at this particular chamber (SIR) throughout the year (Rossi and Lozano, 2016). The modern-day calcite precipitates at or close to isotopic equilibrium, with average $\delta^{18}O$ values of -4.8 $\pm$ 0.3 ‰ (Rossi and Lozano, 2016), whilst the average $\delta^{18}O$ value of stalagmite SIR-14 is -5.2 $\pm$ 0.2 ‰. These values

compare well, suggesting that the oxygen isotopic composition of dripwater was similar to the present day during the early Holocene. Therefore, similar to the present day, the vadose aquifer probably buffered seasonal variations of dripwater $\delta^{18}O$ also during the early Holocene. Additionally, the SIR-14 $\delta^{18}O$ and $\delta^{13}C$ records replicate quite well with the $\delta^{18}O$ and $\delta^{13}C$ records of stalagmite SIR-1 (Rossi et al., 2018: suppl. fig. S5), which is the most reliable indication that they have recorded climate variations, with kinetic or other drip-specific effects playing only a minor role.

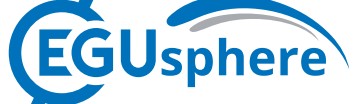



**Figure 4: SIR-14 proxy time-series. From top: $\delta^{18}O$, $\delta^{13}C$, Mg/Ca, Sr/Ca, growth rate and U-Th dates ($2\sigma$ uncertainties). The likely position and duration of the 8.2 ka event are determined from the excursion in the $\delta^{18}O$ signal indicated by the yellow shading. See sect. 5.2.1 for further explanation of the determination for onset/end of the event. The two intervals of relatively low growth rates (GR < 50 mm ka$^{-1}$) are indicated by dotted lines. The variability through these intervals is interpreted to be related to effects specific to the drip site and unrelated to regional climate changes. The growth rate uncertainties ($1\sigma$) are plotted as light blue lines and are derived from the uncertainties in the age-model.**

**Table 1. Correlation coefficients ($r$) for the different proxies ($\delta^{18}O$, $\delta^{13}C$, Mg/Ca, Sr/Ca and growth rate) from stalagmite SIR-14 for the period between 11.7-6.4 ka BP. Correlation coefficients that are statistically significant ($p$-value < 0.05) are marked with an asterisk.**

|  | $\delta^{13}C$ vs $\delta^{18}O$ | $\delta^{13}C$ vs Mg/Ca | $\delta^{13}C$ vs Sr/Ca | $\delta^{13}C$ vs GR | $\delta^{18}O$ vs Mg/Ca | $\delta^{18}O$ vs Sr/Ca | $\delta^{18}O$ vs GR | Mg/Ca vs Sr/Ca | Mg/Ca vs GR | Sr/Ca vs GR |
|---|---|---|---|---|---|---|---|---|---|---|
| correlation coefficient ($r$) | 0.68* | 0.86* | -0.06 | -0.45* | 0.74* | -0.21* | -0.45* | -0.19* | -0.54* | 0.30* |
| sample size (n) | 179 | 179 | 179 | 179 | 179 | 179 | 179 | 179 | 179 | 179 |

## 5 Discussion

### 5.1 SIR-14 proxy interpretation

In the first part of this section, we develop an interpretation of the SIR-14 proxies over the entire record and use this as a basis for interrogating the changes across the 8.2 ka event. We then compare our record of the 8.2 ka event with the timing, duration, structure and interpretation of the event in other proxy records from south-western Europe.

### 5.1.1 Interpretation of growth rate

Speleothem growth-rate variability is controlled by temperature, calcium-ion concentration and/or drip rate (Baker et al., 1998; Genty et al., 2001; Frisia et al., 2003; Borsato et al., 2016). Taken individually, higher levels lead to higher growth rates. Higher temperatures increase bedrock dissolution, leading to higher $Ca^{2+}$ concentrations in the drip waters, and culminating in increased growth rates (Baker et al., 1998; Genty et al., 2001; Frisia et al., 2003). Moisture balance above a cave controls drip rate through its influence on vadose recharge. Increased drip rates deliver more supersaturated solution to the stalagmite, thereby increasing growth rates (Genty et al., 2001; McDermott, 2004).

The growth rate of SIR-14 is relatively high (up to ~220 mm ka$^{-1}$) except for two prominent periods of much slower growth (< 50 mm ka$^{-1}$) at 9.9-9.1 ka BP and 7.9-6.7 ka BP. Through these intervals, the $\delta^{18}O$, $\delta^{13}C$ and Mg/Ca values are higher than for the rest of the record (dotted lines in fig. 4). This convergence of all four proxies suggests a hydrologically driven growth-



rate reduction. During one of these events, growth in the SIR-1 stalagmite stopped altogether (at ~7.7 ka BP: Rossi et al., 2018), supporting the notion that slower growth in SIR-14 at this time could have been caused by a shift to drier conditions.

However, no such evidence is found in SIR-1 to support the same interpretation for the older slow-growth interval in SIR-14 (suppl. fig. S5). This suggests that non-climatic effects specific to the drip site were responsible. The absence of contemporaneous climate changes in other regional palaeorecords during both of SIR-14's slow-growth periods implies that a localised perturbation of infiltration may have affected the two (or more) closely situated source drip points (fig. 2). Thus, the climatic significance of the two slow-growth phases remains equivocal.

Temperature is a potential driver of growth-rate variability in stalagmite SIR-14. Results from a previous study of SIR-1 (Rossi et al., 2018) show low growth rates and high $\delta^{13}C$ values during the Younger Dryas (YD) stadial due to a cooling-induced reduction in soil $CO_2$ productivity. This is supported by poorly developed or absent annual fluorescence laminae during this interval, which point to a temperature-driven reduction of soil organic matter reaching the stalagmite (Rossi et al., 2018). A negative covariation between $\delta^{13}C$ and Mg in SIR-1 at this time, however, also suggests a decrease in effective rainfall.

Temperatures during the Holocene have remained relatively stable, casting doubt on whether temperature changes *per se* would be sufficient to drive growth rate variations in SIR-14. The negative correlations between the growth rate and both Mg/Ca and $\delta^{13}C$ observed in SIR-14 (Table 1), on the other hand, suggest hydrological control.

At El Soplao, most of the rainfall occurs during the winter (suppl. fig. S1: Rodríguez-Arévalo et al., 2011; Ancell and Célis, 2013), and minimal evapotranspiration would accentuate vadose recharge. Three years of drip-water monitoring in the SIR

passage shows relatively stable and high drip rates (> 0.01 s$^{-1}$) throughout the year, in spite of the seasonal positive moisture balance (Rossi and Lozano, 2016). This suggests a strong buffering role of vadose storage, which is not surprising given the thickness of the bedrock overburden (~120 m). Recent reactivation of the drip above SIR-14 indicates that secular changes in drip rate caused by longer-term shifts in the amount of effective recharge, not seen in the modern monitoring study, could explain the modest growth-rate variability in SIR-14.

Across the 8.2 ka event, there is a spike in SIR-14 growth rate, where it increased from ~60 mm ka$^{-1}$ to a maximum of ~220 mm ka$^{-1}$, then back to ~60 mm ka$^{-1}$ (fig. 5). Notwithstanding the large 1σ growth-rate uncertainties for this period, if such an increase were driven only by temperature, it would suggest a *warming* trend through the 8.2 ka event. This runs counter to the proxy evidence across Europe indicating a *cooling* of ~1 ºC through the event (Morill et al., 2013). This strongly implicates hydrological change as the cause the growth rate spike in SIR-14 across the 8.2 ka event.


### 5.1.2 Interpretation of Mg/Ca

Apart from the two prominent peaks that coincide with slow growth rates (fig. 4), the Mg/Ca signal is relatively stable with minor variations throughout the Holocene. The Mg/Ca is positively correlated with $\delta^{13}C$ and $\delta^{18}O$, and negatively correlated with growth rate (Table 1). In stalagmite SIR-1, variations in Mg concentration were related to changes in the Mg/Ca ratio of

the precipitating dripwater, which is mainly controlled by prior calcite precipitation (PCP) and/or water-rock interaction time





(Rossi and Lozano, 2016; Rossi et al., 2018). This interpretation is supported by cave monitoring data in the SIR passage, which shows that Mg/Ca values in the dripwaters are negatively correlated to discharge (Rossi and Lozano, 2016). Additionally, as with SIR-14, the Mg and $\delta^{13}C$ records in SIR-1 are positively correlated, with high values across the YD (Moreno et al., 2010; Baldini et al., 2015; Bartolomé et al., 2015; Baldini et al., 2019) supporting the idea that both speleothem

properties are affected by hydrological changes (Rossi et al., 2018). Thus, it is likely that variations in the Mg/Ca record from SIR-14 are also controlled by PCP and/or water-rock interaction. Both can be attributed to variations in effective recharge linked to rainfall and temperature changes at the surface (Fairchild et al., 2000).

PCP occurs in the cave (e.g. on stalactites or straws above) or in air-filled voids upstream of the cave and removes Ca from the solution before it reaches the stalagmite. With less effective rainfall, the water residence time in the vadose zone increases,

and fractures and voids become dewatered. In addition to potentially enhancing the release of Mg from the bedrock, the resulting degassing and PCP before the waters reach the stalagmite cause Mg/Ca ratios to increase (Fairchild et al., 2000; Fairchild and Baker, 2012). The bulbous shape of the stalactite that is feeding SIR-14 suggests that PCP was the main control. Additionally, the Mg/Ca and $\delta^{13}C$ are positively correlated throughout the whole studied period ($r > 0.8$, Table 1 and suppl. fig. S3), and negatively correlated with growth rate ($r < -0.5$, Table 1 and suppl. fig. S3). Thus, during periods with less

effective recharge the growth rates are low and Mg/Ca and $\delta^{13}C$ values are high in SIR-14 (fig. 4).

It should be noted that variations in the Mg/Ca ratio could potentially also have been influenced by kinetic effects, input from exotic sources of detrital material and marine aerosols, incongruent dissolution, and temperature. Kinetic effects, linked to rapid calcite precipitation, are unlikely because of the relatively low supersaturation observed in modern SIR dripwaters, the high relative humidity in the cave, and the quasi-isotopic-equilibrium conditions evident from cave monitoring (Rossi and

Lozano, 2016). Changes in detrital sources can also contribute to Mg variation. This is unlikely due to the absence of visible detritus-rich laminae, the high $^{230}Th/^{232}Th$ ratios and the high background Mg contributed by the dolostone bedrock (Rossi and Lozano, 2016). Marine aerosols are unlikely to contribute to Mg variability as their influence would be overwhelmed by the high background Mg content of the bedrock and because of the distance from the shore (Rossi et al. 2018). The dolostone host rock can potentially affect the Mg concentrations in the stalagmite through incongruent dolomite dissolution. Due to the lower

solubility of dolomite compared to calcite, $CaCO_3$ is precipitated at the same time as $CaMg(CO_3)_2$ is dissolved. Incongruent dolomite dissolution can be related to increased residence time and PCP, leading to Mg/Ca increases. However, incongruent dolomite dissolution is generally not thought to be a significant process in speleothem-forming environments (Fairchild et al., 2000).

Theoretically, temperature influences the partitioning of Mg into calcite (Katz, 1973; Fairchild and Treble, 2009; Drysdale et

al., 2020) and could exert control on the variations in Mg/Ca in SIR-14. If temperature was the main control, the $D_{Mg}$ would decrease with lower temperatures, causing a reduction in Mg incorporation into the calcite. Based on the pattern of Mg in stalagmite SIR-1 across the end of the YD, an increase in Mg values would be expected, whereas the opposite occurs (Rossi et al., 2018), suggesting temperature has little influence.



The $D_{Mg}$ for SIR-1 is 0.03 ± 0.016 (Rossi and Lozano, 2016; Rossi et al., 2018). The drip above SIR-14 was inactive during
the monitoring period and thus only a first-order estimate of the $D_{Mg}$ of SIR-14 can be calculated, based on the average [Mg/Ca]
of the dripwater in the SIR passage (0.648 ± 0.025 mol mol$^{-1}$) and the average [Mg/Ca] from stalagmite SIR-14 (0.014 ± 0.003
mol mol$^{-1}$). This calculation yields a $D_{Mg}$ in SIR-14 of 0.02 ± 0.005, which is within the range of published values (Huang et
al., 2001; Fairchild et al., 2010; Tremaine and Froelich, 2013). However, this value does not agree very well with the measured
$D_{Mg}$ values in modern stalagmites in El Soplao Cave, which are consistently higher than 0.02, and average 0.03. There are
indications that the drip rate above SIR-14 was relatively fast, suggesting that the paleo-dripwater had a lower Mg/Ca value,
possibly around 0.5 mol mol$^{-1}$. This would give a $D_{Mg}$ of 0.03, which is consistent with the $D_{Mg}$ of modern stalagmites and
SIR-1. The temperature effect on Mg/Ca ratios in SIR-14 can further be discounted due to the similarity in $D_{Mg}$ values between
SIR-14, SIR-1 and modern stalagmites, and the fact that mean annual air temperature has not varied much across the Holocene.
Assuming no major change in the partitioning of Mg into calcite across the Holocene, the Mg/Ca variability in SIR-14 (0.01-
0.03 mol mol$^{-1}$) suggests a Mg/Ca dripwater range of 0.3-0.7 mol mol$^{-1}$. This is similar to the observed variability of [Mg] in
SIR-1 in the early Holocene (0.4-0.7 mol mol$^{-1}$: Rossi et al., 2018), and to the variability of the modern-day dripwaters in the
SIR passage (0.5-0.8 mol mol$^{-1}$; Rossi and Lozano, 2016). This supports the notion that the Mg/Ca variability is controlled by
drip discharge variations in response to changes in recharge.

At around 8.2 ka BP, there is a small decrease in Mg/Ca which aligns with the growth-rate increase, and is similar in amplitude
to other excursions over the preceding ~1000 years (fig. 5). This would suggest that the excursion at 8.2 ka was driven by
enhanced recharge of a magnitude insufficient to induce a prominent Mg/Ca anomaly when viewed in the context of the whole
record.

### 5.1.3 Interpretation of δ$^{13}$C

The δ$^{13}$C is variable through the Holocene section of SIR-14, with general trends similar to the Mg/Ca and δ$^{18}$O profiles (fig.
4). The δ$^{13}$C decreases from the end of the YD through the early Holocene, then shows only minor variations through the
Holocene, with the exception of the two relatively slow growth periods noted above, which are associated with abrupt increases
in the δ$^{13}$C.

The variability of the δ$^{13}$C signal in speleothems is the result of complex interactions that are often challenging to resolve. It
can be influenced by kinetic fractionation which can be induced by rapid $CO_2$ degassing of dripwaters to the cave air or by
evaporation, both causing the δ$^{13}$C values to increase (McDermott, 2004). Kinetic fractionation is unlikely to significantly
affect the δ$^{13}$C in SIR-14 due to the high relative humidity in the cave throughout the year and the near-equilibrium isotopic
conditions (sect. 4.4). Thus, it is likely that the δ$^{13}$C variations in SIR-14 are controlled more directly by external environmental
influences.

In previous work on stalagmite SIR-1, the δ$^{13}$C variability was linked to temperature and hydrological changes above the cave,
with higher values coinciding with cold and dry intervals such as the YD (Rossi et al., 2018). In SIR-14, the δ$^{13}$C is negatively



correlated with growth rate and positively correlated with Mg/Ca (Table 1), thus invoking the possible influence of both temperature and precipitation. Lower temperatures result in reduced biogenic activity in the soil above the cave, which leads to an increase in $\delta^{13}$C values, and *vice versa* (Genty et al., 2003; Genty et al., 2006). However, given the carbon turnover time

in the soil and the relative temperature stability through the Holocene, temperature is unlikely to be the main driver of $\delta^{13}$C variations in SIR-14 through the Holocene. It was established above that growth rate in SIR-14 is most likely controlled by changes in recharge rather than temperature, thus the occurrence of low growth rates and high $\delta^{13}$C values is indicative of decreased drip discharge, which enhances PCP. This is supported by the pattern of proxy changes through the two intervals of slow growth.

Periods with less rainfall can lead to increased PCP, resulting in preferential outgassing of $^{12}CO_2$ from the solution before it reaches the stalagmite, leading to elevated $\delta^{13}$C values (Dreybrodt and Scholz, 2011). The $\delta^{13}$C and Mg/Ca ratio have a consistently high correlation over the studied period ($r > 0.6$: suppl. fig. S3), arguing for a persistent influence of PCP on the $\delta^{13}$C variability. Additionally, monitoring reveals an intra-site correlation between $\delta^{13}$C of modern calcite and the associated dripwater Mg/Ca (Rossi and Lozano, 2016). Thus, this argues strongly for PCP, hence effective recharge, being the main

control on $\delta^{13}$C variability in SIR-14 over the Holocene record.

Across the 8.2 ka interval, the $\delta^{13}$C decreases by ~1‰ between 8.18 and 8.08 ka BP, followed by a reversal of similar magnitude by ~8.06 ka BP. The timing of the excursion is consistent with the growth rate increase and Mg/Ca decrease. However, like Mg/Ca, the $\delta^{13}$C excursion does not stand out as a clear Holocene anomaly, but rather represents one of several excursions of similar magnitude that occur over the preceding ~1000 years (fig. 5). Nevertheless, the correspondence of this

negative $\delta^{13}$C excursion with increased growth rate and decreasing Mg/Ca suggests an interval of increased recharge to the cave.







**Figure 5: SIR-14 proxy time-series from 7.8-9.2 ka BP. From top: δ¹⁸O, δ¹³C, Mg/Ca, Sr/Ca, growth rate (1σ uncertainties) and U-Th dates (2σ uncertainties). The δ¹⁸O displays a well-defined excursion through the event, and its duration is shown by yellow shading. During this interval the δ¹³C and Mg/Ca decrease, but this excursion does not stand out as a clear Holocene anomaly, but rather represents one of several excursions of similar magnitude that occur over the preceding ~1000 years. The Sr/Ca does not show any variation of significance. Across the 8.2 ka event, there is a spike in the growth rate where it reaches a maximum of ~220 mm ka⁻¹. The change in growth rate stands out against the entire record, but it is associated with large uncertainties (reflecting the age model uncertainties).**

### 5.1.4 Interpretation of δ¹⁸O

The SIR-14 δ¹⁸O through the Holocene follows the general trends of the other proxy time series (fig. 4), being positively correlated with δ¹³C and Mg/Ca and negatively correlated with growth rate (Table 1, suppl. fig. S3). The δ¹⁸O decreases from the end of YD through the early Holocene, similar to the δ¹³C, then shows a relatively stable trend, reaching a minimum between 8.5-8.0 ka BP, before a gentle increasing trend until the youngest part of the record (~6 ka BP). Two prominent increases occur during the slow-growth intervals (sect. 5.1.1).

The main drivers of speleothem δ¹⁸O variability are cave temperature and dripwater isotopic composition, provided the calcite is precipitated at or close to isotopic equilibrium (McDermott, 2004; Fairchild et al., 2006; Lachniet, 2009). The stability of mean annual air temperatures during the Holocene would lead to stable cave temperatures, which would implicate changes in dripwater δ¹⁸O as the main driver of δ¹⁸O variability in stalagmite SIR-14. Dripwater δ¹⁸O in temperate regions reflects the weighted mean of local rainfall δ¹⁸O (Baker et al., 2019), which can be controlled by site air temperature, seasonality, rainfall amount, air-mass trajectories and the isotopic composition of the moisture source (McDermott, 2004;). Temperature generally has a direct positive relationship with rainfall δ¹⁸O in the middle to high latitudes (Dansgaard, 1964). In northern Spain, this gradient is between +0.24 and +0.34 ‰ ºC⁻¹ (Domínguez-Villar et al., 2008). However, it is counterbalanced by the temperature dependence of water-calcite isotope fractionation in the cave (-0.23 ‰ ºC⁻¹: Kim and O'Neil, 1997). Thus, the net effect of a 1ºC decrease in temperature, as purported to have occurred during the 8.2 ka event (Morill et al., 2013), is an increase in local speleothem δ¹⁸O of between 0.01 and 0.11 ‰ ºC⁻¹, which is clearly exceeded in the SIR-14 record (fig. 4). Other factors affecting rainfall δ¹⁸O must be more important.

Meteorological data from the nearby Santander GNIP station (2000-2015 AD) show an annual cycle in the δ¹⁸O of the rainfall, with mean summer (May-September) values of -3.71 ± 1.35 ‰ (VSMOW) compared to -6.69 ± 1.43 ‰ (VSMOW) in winter (October-March) (suppl. fig. S1: Rodríguez-Arévalo et al., 2011). The monthly δ¹⁸O values of rainfall are positively correlated with monthly temperatures ($r = 0.6$, $p < 0.001$, gradient of 0.28 ºC⁻¹) and negatively correlated with the monthly rainfall amount ($r = -0.5$, $p < 0.001$) in the region. However, deconvolution of these data show that neither the rainfall amount nor temperature exerts a *persistent* influence on rainfall δ¹⁸O (suppl. fig. S2), highlighting the complexity in interpreting regional rainfall δ¹⁸O values, and consequently, speleothem δ¹⁸O values.



The temperature effect on rainfall δ¹⁸O can occur via changes in the seasonal distribution of rainfall amount (Denton et al., 2005; Lachniet, 2009). A significant change in seasonality, such as a shift from winter-dominant to summer-dominant rainfall, would affect speleothem δ¹⁸O because it changes the annual amount-weighted rainfall δ¹⁸O, which is transmitted to the

dripwater δ¹⁸O. Monitoring of the Soplao Cave dripwater δ¹⁸O shows mean values of -6.70 ± 0.20 ‰ (VSMOW) (Rossi and Lozano, 2016), which is consistent with the isotopic composition of winter rainfall in the region (Rodríguez-Arévalo et al., 2011). The δ¹⁸O values of individual drips in the cave show insignificant seasonal variations (~0.1 ‰), and the range of values throughout the cave is narrow (from -7.08 ‰ to -6.27 ‰ VSMOW: Rossi and Lozano, 2016). Additionally, no systematic variability between winter and summer drip samples is observed, suggesting that the infiltrated rainwater mainly reflects the

predominant winter recharge and is homogenised in the vadose zone (Rossi and Lozano, 2016). Whilst there is no seasonal variability in the modern-day dripwater composition over the short-term, long-term variations in seasonal distribution of precipitation could have affected the variability of SIR-14 δ¹⁸O in the past. This effect has been suggested in a study from nearby La Garma Cave, where a rainfall model was used to extract information about seasonal shifts in rainfall and temperature in the past (Baldini et al., 2019). This model indicates that wet winters and dry summers could have occurred for 71 % of the

Holocene; there were also periods that could have been dominated by dry winters and wet summers (Baldini et al., 2019).

The positive correlation between δ¹⁸O and both δ¹³C and Mg/Ca and the negative correlation between δ¹⁸O and growth rate implies that changes in the amount of recharge reaching the cave was the main driver for δ¹⁸O variability. Whether this reflects changes in total annual rainfall amount or changes in the seasonal distribution of rainfall is difficult to unravel.

The notion that the δ¹⁸O in SIR-14 is mostly influenced by rainfall amount is supported by the present-day relationship between

rainfall amount and the δ¹⁸O of rainfall in the region (Rodríguez-Arévalo et al., 2011). The variability of δ¹⁸O in SIR-1 has been interpreted to reflect changes in the rainfall amount (Rossi et al., 2018). This interpretation is supported by other stalagmites in the region (Cueva de Asiul: Smith et al., 2016; Kaite Cave: Domínguez-Villar et al., 2008). However, as alluded to above (supp. fig. S2), and pointed out previously (Baldini et al., 2019), the apparent causation behind the present-day relationship between rainfall δ¹⁸O and rainfall amount may be a statistical artefact. It is not possible to rule out the role of

seasonality on rainfall δ¹⁸O variability. An increase in the ratio of summer-to-winter rainfall, without a net change in annual amount, would induce a higher amount-weighted mean δ¹⁸O, which would be exacerbated by higher summer evaporation, resulting in less effective recharge to the cave. This would leave its mark on (higher) δ¹³C and Mg/Ca values and (lower) growth rate, having the same effect on the proxies as a reduction in annual rainfall amount. Thus, although changes in rainfall amount have been argued by nearby studies as the main driver of speleothem δ¹⁸O, the role of seasonality cannot be overlooked.

Thus, translating the effect of changes in recharge on dripwater (and speleothem) δ¹⁸O into the climatically more meaningful terms of rainfall amount or rainfall seasonality remains a conundrum. Thus, we restrict our interpretation of SIR-14 δ¹⁸O variations more generally to effective recharge.

The correlation between δ¹⁸O and the other proxies varies through time (suppl. fig. S3), especially between 9 and 8 ka BP, when the correlation is weaker. Within this time interval, the 8.2 ka event is recorded as a prominent multi-pronged decrease



in $\delta^{18}O$ (fig. 5). This aligns broadly with the pattern of $\delta^{13}C$, Mg/Ca and growth rate changes, but the $\delta^{18}O$ decrease appears to lead the other proxies. One possibility for this is a change in the isotopic composition of the moisture source.

At the time of the 8.2 ka event, large volumes of ice-sheet meltwaters with drastically lower $\delta^{18}O$ values than those of the surrounding ocean entered the North Atlantic (Barber et al., 1999). In spite of ocean mixing, this resulted in a change in surface water isotopic composition which could have altered rainfall $\delta^{18}O$ values around the North Atlantic basin. Model simulations of the event predict a ~2 ‰ decrease in the $\delta^{18}O$ of the rainfall over Greenland and a ~1 ‰ decrease in $\delta^{18}O$ of the rainfall over western Europe (Tindall and Valdes, 2011). The meltwater fluxes have been associated with a decrease in $\delta^{18}O$ values in planktic foraminifera (e.g. Ellison et al., 2006) and a decrease in speleothem $\delta^{18}O$ values in nearby Kaite Cave (Domínguez-Villar et al., 2009) (fig.1). Thus, it is likely that the negative $\delta^{18}O$ values in SIR-14 across the 8.2 ka event are indeed recording the freshwater fluxes released to the North Atlantic during the event, causing a modification in the local isotopic composition of precipitation.

### 5.1.5 Synthesis

To summarise, the SIR-14 proxies record hydrological changes through the early Holocene. In periods with more effective recharge, the $\delta^{13}C$ and Mg/Ca are low and the growth rate is high, and *vice versa* when recharge is lower. The exception is for the two periods of relatively slow growth, in which drip rate decreased due to a local effect in the drip that fed SIR-14 (as discussed in sect. 5.1.1). Understanding the variation in the $\delta^{18}O$ signal in stalagmite SIR-14 is challenging, as this signal is probably controlled by a combination of factors. Throughout the Holocene, the $\delta^{18}O$ was also influenced by effective recharge, but decoupling the effects of rainfall amount and/or seasonality remains elusive. Nonetheless, lower values indicate more recharge (either more annual rainfall or more winter rainfall) and *vice versa* for higher values. During the time interval of the 8.2 ka event, there is a multi-pronged centennial-scale negative $\delta^{18}O$ excursion, starting at $8.19 \pm 0.06$ ka BP and lasting until $8.05 \pm 0.05$ ka BP (fig. 5). The commencement of this excursion precedes the changes in the other proxies, implicating a source-water effect associated with the meltwater fluxes to the North Atlantic at the time. The fact that the other proxies display evidence for enhanced cave recharge suggests that part of the $\delta^{18}O$ change during the 8.2 ka interval may be related to local water balance.

Importantly, the changes in all geochemical proxies across SIR-14 are not 'stand-out' features of the entire record, unlike the case in NGRIP and some European records (e.g. Ammersee: von Grafenstein et al., 1999). Only the growth-rate change stands out against the entire record, but this is associated with large uncertainties (reflecting the age-model uncertainties). The prominence of the growth-rate change is also conditioned by the higher-density of U-Th dating, which can produce short-term changes in the age-depth model. The apparent complacency of the response to the 8.2 ka event in SIR-14 suggests that the event either produced minimal impact on the climate of this part of south-western Europe or was not strong enough to induce large changes in the geochemical proxies in this particular cave system.





### 5.2 The 8.2 ka event in the Iberian Peninsula

#### 5.2.1 Timing and duration

It is widely accepted that the drainage of glacial lakes Agassiz and Ojibway caused perturbation of the AMOC and triggered the 8.2 ka event (Barber et al., 1999; Li et al., 2012; Törnqvist and Hijma, 2012). However, due to disagreement between continental and marine records, the timing, duration and number of freshwater fluxes remain debated (e.g. Carlson and Clark, 2012; Brouard et al., 2021). The most common explanation for the trigger of the event has been one large outburst of freshwater from lakes Agassiz and Ojibway, as documented by a red marker bed in Hudson Strait sediment cores and dated to $8.47 \pm 0.3$

cal ka BP (Barber et al., 1999). This has been supported by several other studies of red marker beds in the region (Lajeunesse and St-Onge, 2008; Jennings et al., 2015; Lochte et al., 2019). However, several records show evidence for multiple meltwater fluxes prior to the event (e.g. Ellison et al., 2006; Hillaire-Marcel et al., 2007; Lochte et al., 2019), arguing for a more complex drainage history. A recent study proposes that the 8.2 ka event was triggered by two distinct drainage events: a subglacial drainage at $8.22 \pm 0.02$ cal ka BP and the final drainage of LAO at $8.16 \pm 0.02$ cal ka BP (Brouard et al., 2021). The timing of

the first meltwater pulse is based on an annual chronology from well-defined varve records, whilst the timing of the second meltwater pulse was determined by using the minimum- and maximum-limiting $^{14}$C ages of continental records that showed clear evidence of the termination of glaciolacustrine conditions (Brouard et al., 2021). The subglacial drainage coincides with the onset of the 8.2 ka event in the $\delta^{18}$O in the NGRIP ice core ($8.25 \pm 0.05$ ka BP: Thomas et al., 2007) and the final drainage appears in the central part of the event, coinciding with the lowest $\delta^{18}$O values in the NGRIP ice core (fig. 6). However, it

should be noted that the timing of the subglacial drainage is based on varve counting between two sediment units, so the final drainage may have eroded some of the varves, effectively making the age too young.

Since a focus of this paper is the timing and synchronicity of the event between Greenland and terrestrial archives in south-western Europe, we need to consider discrepancies in timing, which are often explained by poorly constrained chronologies and/or low-resolution time series. Thus, to circumvent this we explore the synchrony of the 8.2 ka event by selecting south-

western Europe palaeorecords that meet the following criteria: 1) continuous temporal coverage across the 8.2 ka event (i.e. 8.5-7.9 ka BP); 2) a minimum temporal resolution of 40 years per data point between 8.5-7.9 ka BP; and 3) unambiguous interpretation of the proxies (as indicated by the authors). The palaeorecords have been sourced from the SISAL database (Atsawawaranunt et al., 2018; Comas-Bru et al., 2019; Comas-Bru et al., 2020) or through personal communication with the authors. There are several high-quality lake archives in south-western Europe. However, none has a high enough resolution or

sensitivity to explore the timing of the 8.2 ka event (Morellón et al., 2018). Multiple speleothem records cover the period around 8.2 ka in south-western Europe. However, only five meet the selected criteria (Table 2; fig. 6). For this reason, only speleothem records (and the NGRIP ice core) will be used in the comparison of the timing and synchrony. The selected records were standardised following the procedures of Corrick et al. (2020), as summarised here. First, to avoid potential biases in age-depth modelling methodology, the chronology of each time series was revised and a new age-depth model was constructed

(suppl. fig. S7). All the ages were recalculated using the most recent estimates of the decay constants from Cheng et al. (2013).



Estimates of the initial $^{230}$Th/$^{232}$Th activity ratios were revised using one of several potential methods (Hellstrom, 2006), where the most common method to correct for detrital thorium assumes a bulk Earth $^{230}$Th/$^{232}$Th activity ratio of $0.82 \pm 0.42$. However, this value does not cover the full variability of $^{230}$Th/$^{232}$Th in speleothems (Hellstrom, 2006). Where the bulk Earth value was assumed, a $^{230}$Th/$^{232}$Th activity ratio of $1.5 \pm 1.5$ has been used instead, as it has been shown to be a better estimate

given the range of values observed in speleothems (Hellstrom, 2006). In some cases, the stratigraphic constraint approach (Hellstrom, 2006) was used to refine the correction to a speleothem-specific estimate. Where lamina counts were used to construct a chronology, the published chronology was retained (e.g. the chronology of the LV5 record from Kaite Cave: Domínguez-Villar et al., 2017).

In attempting to identify the 8.2 ka event, we followed the approach of Thomas et al. (2007) for the onset and end of the event.

In the Greenland ice cores, the onset (end) is defined as the first (last) isotopic data point below the baseline that precedes the event (suppl. fig. S6). The baseline is calculated based on the mean of the preceding 1000 years (between 9.3-8.3 ka BP). However, when this method was followed strictly, the position of the isotopic data point of the onset and end (i.e. peak, mid-point, trough) varied between different archives due to variable resolution and signal variability. Additionally, the Holocene signal-to-noise ratio in speleothems is low and applying a statistical approach to strictly define the onset/end of the event was

found inappropriate in some cases. Thus, to ensure that the comparison of the onset and end of the event is based on consistent positioning between the archives, it was necessary to shift some of the isotopic points to positions that are structurally similar to that of the Greenland ice core, irrespective of their timing (suppl. fig. S6).









**Figure 6: Comparison of δ¹⁸O records across the 8.2 ka event. From top: Timing of the drainage of Lake Agassiz and Ojibway (LAO) (Barber et al., 1999; Buoard et al., 2021); NGRIP ice core record (North Grip Ice Core Project Members, 2004); speleothem records of SIR-14 (this study), GARO1 (La Garma Cave; Baldini et al., 2019), LV5 (Kaite Cave; Domínguez-Villar et al., 2009), GdL2016-1 (Galeria das Lâminas; Benson et al., 2021), Cha-1 (Chaara Cave: Ait Brahim et al., 2019). The intervals of slow growth in SIR-14 (9.9-9.1 ka BP and 7.9-6.7 ka BP) are shown as dotted lines as they are representative of drip-specific effect and not regional climate variability. The duration of the event from the NGRIP record is shown by yellow shading. The onset (red) and end (yellow) of the event are indicated by coloured markers (see text for explanation for determining the onset and end of the event).**

**Table 2. The timing of the 8.2 ka event from the selected archives discussed in this paper. See text for explanation for determining the onset and end of the event.**

| Site | Event start ± 2$\sigma$ (year BP) | Event end ± 2$\sigma$ (year BP) | Event duration ± 2$\sigma$ (years) |
|---|---|---|---|
| NGRIP (Greenland) | 8,250 ± 49 | 8,100 ± 46 | 150 ± 67 |
| El Soplao Cave (Spain) | 8,191 ± 62 | 8,054 ± 54 | 137 ± 83 |
| La Garma Cave (Spain) | 8,186 ± 208 | 8,027 ± 139 | 159 ± 250 |
| Kaite Cave (Spain) | 8,213 ± 62 | 8,146 ± 62 | 67 ± 88 |
| Galeria das Lâminas (Portugal) | 8,234 ± 181 | 8,144 ± 156 | 90 ± 239 |
| Chaara Cave (Morocco) | 8,326 ± 110 | 8,204 ± 91 | 122 ± 143 |

In the Greenland ice cores, the event started at 8.25 ± 0.05 ka BP and lasted until 8.09 ± 0.05 ka BP, giving a total duration of 160 years (Thomas et al., 2007; yellow shading in fig. 6). In stalagmite SIR-14, the event is manifested as a negative excursion in the δ¹⁸O, starting at 8.19 ± 0.06 ka BP and lasting until 8.05 ± 0.05 ka BP. This is statistically indistinguishable from the timing of the event in Greenland (fig. 6). The other archives also overlap with this timing within age uncertainties. This suggests that the event was indeed synchronous between Greenland and south-western Europe.

To statistically verify this claim, we performed a chi-square test (also known as the mean square weighted deviation, MSWD) implemented in Isoplot (Ludwig, 2012) for the selected archives. This test shows all ages for the onset of the event to be statistically from the same population (MSWD = 1.16, $p$ = 0.33), yielding an error-weighted mean age of 8.23 ± 0.03 ka BP, suggesting that the onset of the event was synchronous. Performing the test on all the ages from the end of the event gives a low probability of fit ($p$ = 0.04), suggesting that the pool of ages is not from the same population. The Chaara Cave record has the least fitting age in the population. For this record it was deemed inappropriate to use the statistical method of Thomas et al. (2007). The position of the isotopic point for the end of the event was shifted to a structurally similar position to that of the NGRIP record (suppl. fig. S6). However, the recovering phase of the event and the following decades are more gentle than in the other archives, hence there could be other explanations for this discrepancy as well. Nevertheless, when we exclude this age from the chi-square test for the end of the event, all ages are statistically from the same population (MSWD = 1.6, $p$ = 0.17), resulting in an error-weighted mean age of 8.10 ± 0.05 ka BP, suggesting a synchronous end of the event for five of the selected records.



### 5.2.2 Structure

The 8.2 ka event in the Greenland ice cores is asymmetric in shape and preserves evidence of decadal variability (Thomas et al., 2007). The onset of the event is relatively abrupt, after which there is an interval of fluctuating values that are consistently below the mean of the preceding 1000 years. Within this interval there is a double-trough, which includes the lowest $\delta^{18}O_{ice}$ values across the event. Finally, there is a multi-stage recovery to pre-event $\delta^{18}O_{ice}$ values, including an apparent overshoot at the end of the recovery phase (North Grip Ice Core Project Members, 2004; Thomas et al., 2007). Some models have reproduced the asymmetric shape and attributed it to a sudden weakening of the AMOC (e.g. Renssen et al., 2001), with two peaks attributed to a strengthening or temporary recovery of the AMOC ~30 years after the initial freshwater perturbation (Wiersma and Renssen, 2006).

The selected speleothem records all show a decrease in $\delta^{18}O$ values across the event and similar structures compared to the Greenland ice-core record (fig. 7). For instance, the sharp onset, double-trough and multi-stage recovery phase are visible to some degree in all the records, although there are substantial differences in detail at decadal timescales, which are difficult to reconcile owing to age-model uncertainties and the variable resolution. The double trough is marked by at least two (and sometimes more) minima in $\delta^{18}O$ values during the event, and is recognisable in all the selected records. These minima in $\delta^{18}O$ could support the hypothesis of two or more meltwater fluxes as proposed by several authors (e.g. Godbout et al., 2019; Brouard et al. (2021).

Another often-discussed feature of the 8.2 ka event is the 'precursor event', as reported in the Kaite Cave record (fig. 7) (Domínguez-Villar et al., 2009). This strong negative anomaly in the Kaite Cave $\delta^{18}O$ record, centred at 8.35 ka BP, was interpreted as reflecting an early meltwater pulse to the North Atlantic (Domínguez-Villar et al., 2009). It has been reported in both continental and marine archives in the Hudson Bay – Labrador Strait region (Hillaire-Marcel et al., 2007; Lajeunesse and St-Onge, 2008), in the North Atlantic (Ellison et al., 2006) and in stalagmite records from the Iberian Peninsula (Garma Cave: Baldini et al., 2019 and Galeria das Lâminas: Benson et al., 2021). However, it is less obvious and of lower amplitude in these two stalagmite profiles compared to the strong anomaly in Kaite Cave. In the SIR-14 speleothem $\delta^{18}O$ record, there is no clear indication of a precursor event, similar to the Greenland ice cores (fig. 7).

The varve-dated Mondsee $\delta^{18}O$ record in Austria displays higher-than-average $\delta^{18}O$ values during the first decades after the 8.2 ka event (Andersen et al., 2017), which the authors relate to an $\delta^{18}O$ overshoot in response to regional climate changes from the AMOC recovery. This feature can also be observed in SIR-14 (fig. 7: blue shading), but is not clearly visible in the other selected speleothem records used in this comparison.







**Figure 7: Comparison of δ¹⁸O variability from selected archives across the 8.2 ka event. From top: NGRIP, SIR-14, GAR01, LV5, GdL2016-1, Cha-1 (for references, see fig. 5). The duration of the 8.2 ka event (as inferred from the NGRIP record) is highlighted by yellow shading. The double trough (central part of the event; Thomas et al. 2007) is marked with a grey, thick line in the NGRIP**
**profile. Orange shading indicates the "precursor event" identified in the Kaite Cave record. Blue shading indicates the interval where δ¹⁸O overshoots after the event in the NGRIP and SIR-14 records.**

### 5.2.3 Impact

From stalagmite SIR-14, the 8.2 ka event is shown as a well-defined negative excursion in δ¹⁸O, interpreted to reflect, at least
in part, a change in the isotopic composition of the ocean source water due to freshening of the North Atlantic by ice sheet meltwaters. Nevertheless, across the Holocene, changes in recharge appear to be the main control on speleothem δ¹⁸O, with either more annual rainfall and/or more winter rainfall leading to lower δ¹⁸O values, and *vice versa*. These observations in SIR-14 are consistent with the speleothem record from Kaite Cave, where Holocene variability is controlled mainly by variations in precipitation, but where the δ¹⁸O variation during the 8.2 ka event reflects changes in North Atlantic surface water δ¹⁸O
(Domínguez-Villar et al., 2008; Domínguez-Villar et al., 2009). The other proxies measured in SIR-14 also capture the event, with δ¹³C and Mg/Ca values decreasing and growth rate increasing (fig. 5), suggesting enhanced recharge at the time, but these changes are not outstanding in the context of the Holocene, and we suggest SIR-14 records only a minor climate impact from the 8.2 ka event.

The climatic impact across the 8.2 ka event has been interpreted as a hydrological change in several other palaeorecords from
the region. For instance, North Atlantic cold episodes are associated with wetter conditions in Chaara Cave in northern Morocco based on two δ¹⁸O speleothem records from the same cave (Ait Brahim et al., 2019). Speleothem petrography combined with isotopic changes from Cova da Arcoia in Northern Spain support the connection between Atlantic cold episodes and wetter conditions (Railsback et al., 2011). Speleothem δ¹⁸O record from Villars Cave in France has been interpreted to reflect increased rainfall during the event (Ruan, 2016). In La Garma Cave, the variability of speleothem δ¹⁸O has been
interpreted to be controlled by seasonal changes in temperature and rainfall via a model. The model suggests increased winter rainfall and a significant temperature decrease (1.7 ºC lower than the early Holocene average), shown as a decrease in δ¹⁸O, in northern Spain during the 8.2 ka event (Baldini et al., 2019). In Portugal, a slightly different picture emerges: growth-rate variability over the Holocene is determined to be a proxy for winter precipitation changes, with wetter conditions associated with cold Atlantic episodes and negative NAO phases (Benson et al., 2021). The δ¹⁸O from the same record shows a clear
negative excursion through the 8.2 ka event (fig. 7), but its relationship to winter rainfall amount is unclear because there is no concurrent increase in growth rate change at the time. Thus, it could be argued that the δ¹⁸O excursion in this stalagmite is also reflecting mostly a change in the moisture source δ¹⁸O associated with the 8.2 ka event.

From the region's speleothem palaeorecords, there are two main conclusions from the interpretation of the δ¹⁸O excursion across the 8.2 ka event: increased winter rainfall (e.g. Baldini et al., 2019) and change of the moisture source composition
associated with the discharge of freshwater into the North Atlantic Ocean by the outburst(s) of Lake Agassiz-Ojibway





(Domínguez-Villar et al., 2009). The SIR-14 proxy data support these previous conclusions, nonetheless with a slight caution about the seasonality of recharge increase, which was not possible to associate firmly to annual rainfall or winter rainfall. The SIR-14 $\delta^{18}$O record has been interpreted to reflect, at least in part, a change in the isotopic composition of the ocean source water, supporting the latter conclusion (e.g. Domínguez-Villar et al., 2009).

There are numerous other terrestrial palaeoarchives in south-western Europe providing important information about palaeoclimate across the Holocene (e.g. Pèlachs et al., 2011; Morellón et al., 2018; Zielhofer et al., 2019), but the brevity of the 8.2 ka event, the low resolution and/or insufficient age control of these records precludes the drawing of meaningful conclusions regarding the event (Morellón et al., 2018).

## 6. Conclusions

The SIR-14 stalagmite records hydrological changes through the early Holocene in northern Spain, as determined from $\delta^{18}$O, $\delta^{13}$C, Mg/Ca and growth-rate properties. Variability in each is mainly driven by changes in effective recharge. Periods of enhanced recharge occur as lower $\delta^{13}$C, $\delta^{18}$O and Mg/Ca values, and higher growth rates, and *vice versa*. Whether changes in winter rainfall or annual rainfall amount drove these changes cannot be determined. During the 8.2 ka event, the $\delta^{18}$O is influenced by a change in the moisture source isotopic composition due to fluxes of meltwater to the North Atlantic from lakes

Agassiz and Ojibway. Change in the other proxies at the time suggest enhanced recharge as well, but are within the range of typical variability over the preceding 1000 years, perhaps with the exception of growth rate. This suggests that the 8.2 ka event had a minor climate impact at our site.

Our results are broadly consistent with other speleothem records of the 8.2 ka event from the region, where two explanations for lower $\delta^{18}$O values are proposed: increased winter rainfall (as shown in the Garma Cave record (Baldini et al., 2019) and

supported by the Galeria das Lâminas record (Benson et al., 2021)), and a modification of the isotopic composition of the moisture source (as shown in the Kaite Cave record (Domínguez-Villars et al., 2009).

Based on the $\delta^{18}$O record in SIR-14, the 8.2 ka event started at 8.19 ± 0.06 ka BP and lasted until 8.05 ± 0.05 ka BP. This is slightly delayed compared to the NGRIP ice core record, but within the combined uncertainties of both records. By using a few selected high-quality and high-resolution records, we have shown that the event is statistically synchronous between

Greenland, south-western Europe and northern Morocco, giving a regional error-weighted mean age for the onset of 8.23 ± 0.03 ka BP and 8.10 ± 0.05 ka BP for its completion. The structures of the event in the Greenland ice cores can be recognised in SIR-14, including the sharp onset, double trough, multi-stage recovery phase and the $\delta^{18}$O overshoot after the event. Some of these features can also be observed in the other selected archives. However, no precursor event was identified, despite its purported recognition in other regional records.


**Data availability.** The SIR-14 proxies and U-Th results from this study will be made available through the SISAL database.



**Author contribution.** I. Couchoud and R. Drysdale conceived the original study. C. Rossi provided sample material and
conducted the field work. H. Kilhavn carried out the U-Th measurements and the carbon and oxygen isotope analyses. J.
Hellstrom conducted all the U-Th analyses, produced the age models for SIR-14 and SIR-1, and reviewed the age-depth models
of published studies. H. Wong carried out the trace element analyses. H. Kilhavn conducted the initial data interpretations,
created the figures and wrote the first draft of the manuscript, with input from I. Couchoud and R. Drysdale. These were
elaborated upon by the other authors. All authors reviewed and edited the final manuscript.

**Competing interests**. The authors declare that they have no conflict of interest.

**Acknowledgements.** We wish to thank the authors who made their data available either upon request or by uploading them to
online repositories. H. Kilhavn beneficiated of a PhD grant funded by the MOPGA (Make Our Planet Great Again) program
and USMB (Université Savoie Mont Blanc). This work was supported by the French National program LEFE (Les Envelopes
Fluides et l'Environnement). C. Rossi acknowledges the financial support received from project RTI2018-094155 B-100
(MCI/AEI/FEDER, UE). Petra Bajo and Ellen Corrick are thanked for assisting with U-Th dating at the University of
Melbourne. R. Drysdale acknowledges the financial support of the Australia Nuclear Science and Technology Organisation
for the trace element analyses.

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
