# Peer review of "The 8.2 ka event in northern Spain: timing, structure and climatic impact from a multi-proxy speleothem record"

_EGUsphere, 2022_

## Author Comment (AC1)

**General comments**

- Multiple records suggest that the 8.2 ky event is the most significant climate anomaly of the Holocene. It is likely to have been triggered by melt waters in the North Atlantic region allowing us to examine a climate mechanism that may well operate in the near future. Thus, the authors certainly address a relevant subject.
- The authors select a cave from the North Atlantic region, close to the source of the perturbation.
- They provide a reconstruction of the hydroclimate of the regions using multiple proxies and with decent age control. They make a careful examination of the proxy interpretation within the record they produce, pointing out strengths and limitations.
- They further compare this record to others in the region providing regional context to the event.
- I think the goals of the project are highly relevant. My major comments address one technical calculation aspect and consideration of a recently published paper that is relevant to the conclusions in this paper.

We thank reviewer 1 for these very positive and constructive comments and suggestions.

**Specific comments**

- Growth rate calculations:
  - One of the strengths of using speleothems as a proxy archive is the strong age control. This is provided by absolute uranium-thorium dating with well-constrained uncertainties. Conversion of these absolute ages to an age-depth model has to incorporate uncertainties resulting from sampling resolution, averaging of time during sampling and other statistical considerations that the age-depth model may make.
  - I think a more robust way to go about growth rate calculations is to measure the growth rate between uranium-thorium dates, rather than stable isotope and trace element sample depths.
  - This study has made sufficient age measurements to make this approach feasible.
  - To examine the relationship between growth rate and stable isotopes and trace elements, I would then average these proxy measurements between uranium-thorium depth samples.
  - I would highly recommend that the authors carry out this exercise, if not as the main analysis, then at least for verification purposes.

We thank reviewer 1 for this suggestion. Below we compare the growth rate derived from our depth-age model (Finite Positive Growth Rate Model, FPGRM, in blue) and the growth rate derived directly between the U-Th dates (plotted on the average age value in red and plotted between the dates in black). The comparison shows that both approaches are feasible. The FPGRM is based on Monte-Carlo Bayesian statistics and involved 10,000 simulations. For each iteration, each age determination is randomised within its uncertainty (Corrick et al., 2020). Amongst other things, the reviewer is suggesting that an ADM must incorporate depth uncertainty: this is exactly what the FPGRM does, using the depth uncertainties listed in the U-Th data table. Thus, the depth uncertainty is fully propagated into the age model simulations.

The depth-age model is created by a least-squares best-fit age sequence that is fitted through the U-Th dates and their uncertainties. The FPGRM also interpolates the uncertainty between two age determinations, permitting the growth rate to vary over an order of magnitude at all time scales (Corrick et al., 2020).

The alternative procedure suggested by the reviewer - where growth rate is derived directly and linearly between each pair of U-Th dates - does not take into account the fact that the uncertainties of the individual U-Th ages help constrain the path of the median ADM, meaning that, generally, the median ADM will more likely intersect ages with small measurement uncertainties compared to ages with large measurement uncertainties, based on the principles of probability. Furthermore, the approach will fail where there is an age inversion. Consider a pair of consecutive, plausible U-Th ages: a stratigraphically older age of 8.26 ± 0.13 ka succeeded by a stratigraphically younger age of 8.29 ± 0.15 ka. Using the reviewer's approach would lead to a negative growth rate between these two dates, whereas the FPGRM recognises that the ages overlap within error and would construct an age model that guarantees a positive growth rate (as should be the case).

The figure below shows that the different methods compare well. The most significant difference is shown in the larger, more abrupt variability in the growth rate derived directly between the U-Th dates (suggested by the reviewer), which leads to more unrealistic instantaneous jumps in growth rate. We suggest that including the age- and depth uncertainties via the FPGRM, and allowing the growth rate to vary at all time scales, produces more realistic growth rate time series.

The growth rate derived from the FPGRM was interpolated to the stable isotope depths in the manuscript (figure 4 and 5) in order to directly correlate growth rate versus stable isotopes and trace elements. We used the lowest resolution proxy (i.e., the trace elements) for this comparison and correlation analysis.

[Figure]

- Timing and structure:
  - The recently published paper by Parker and Harrison on 'examination of the timing, duration and magnitude of the 8.2 ka event in global speleothem records' would be a useful reference to contextualize the region examined in this study to global records in the Parker and Harrison study.

We thank reviewer 1 for pointing out this important paper. We have added reference to this paper in the discussion on the timing of the 8.2 ka event (see section 5.2.1)

- More nuanced characterization of the trigger region:
  - It is often the case that the driver of a climate events, such as the 8.2 ky event, is 'found' in distal locations such as the monsoon regions. This is not surprising, but nevertheless, it would be nice if the event could be better characterized at the source location.
  - The authors have a sample and experience with the climate of such a trigger source location. Where possible, it would be nice to see if the authors could discuss their results against modeled data and data from other archives and proxies in the region to give a better picture of the event. This would help understand the climate dynamics in other locations as well and would make the study more useful and impactful.

We thank reviewer 1 for this suggestion. It is true that the 8.2 ka event is found in distal locations, including in the monsoon regions. This is likely due to perturbations of the AMOC leading to a shift in the position of the Inter Tropical Convergence Zone (ITCZ), which controls the strength of the monsoon. From our site we show that the event is capturing the meltwater fluxes, as it comes under the direct influence of the North Atlantic. However, as discussed in the manuscript, the event in our record is not shown as a major excursion in the various proxies, which constrains our interpretation and characterization as it is described now.

We agree with the reviewer that a climate model would improve the understanding of climate dynamics in other locations as well. However, this is beyond the scope of our paper but is the focus of another part of the lead author's PhD project, which looks at the 8.2 ka event across Europe comparing proxy and model data. We have added some discussion regarding other records in the region, as we agree with the reviewer that they offer important information about the climate in the early Holocene (see reply to reviewer 2 and section 5.2.3).

**Specific comments**

Line 45: Does the ice layer counting effect the start and end age or also the duration of the event?

The counting uncertainty for the NGRIP ice-core record is estimated to be ± 47 years for both the start and end of the event, based on the GICC05 age-model. This implies that, whilst the start and end have ± 47 year uncertainty, the duration of the event has zero uncertainty.

Line 55: Please can you add references for 'event, duration, shape and impacts' as well as the attribution to 'low resolution time series'.

We are not sure why references are needed here. Perhaps the reviewer can elaborate?

Stay consistent with units of trace element measurements through the paper.

We have checked the manuscript and modified the units to be consistent throughout the paper (all trace element measurements are given in mmol mol$^{-1}$).

Figure 1: Show Cantabrian region and the Santander GNIP locations on the map.

Figure 1 has been modified to include the location of the Santander GNIP station and the Cantabrian region.

Is it monthly averaged d18O or rainfall-weighted monthly averaged d18O?

The d18O values of the rainfall from the Santander GNIP station have been updated, they are now plotted as monthly amount-weighted d18O values.

Figure 2: Thanks for the photos! It would be great if you could provide some additional information. E.g. what is the height of the drip water from the stalagmite? Do you know if the water is dripping through the stalactite, or if it is blocked, and flowing outside the stalactite.

The current drip height (i.e. from the tip of the stalactite to the top of the stalagmite) is 37 cm, and the total length of the stalactite is 45 cm. Currently, the water appears to flow through the central canal of the stalactite, as the outer surface is relatively dry, and the very tip of the stalactite is shaped as a soda straw. However, the middle part of the stalactite has a bulbous shape, suggesting that, in the past, the water could have also been flowing over the outer surface of the stalactite.

Line 110: Rainfall 'amount effect' is a rather technical term used to describe an isotopic process more relevant to tropical convective systems. Perhaps this is rather upstream rainout?

Rainfall amount effect is indeed most evident where deep tropical convection processes occur. However, it has also been observed in extratropical regions (e.g. Treble et al. 2005 EPSL for the SW Western Australia, and Drysdale et al. 2020 Nat. Comms for Corchia Cave, Italy). A test of the presence of the rainfall amount at a site can be evaluated using paired monthly rainfall amount and rainfall d18O data, as we do for the Santander GNIP station data. By controlling for monthly site temperature, we show that the rainfall amount effect is an unimportant phenomenon at this site.

Paragraph starting at 130: It's great that so much cave exploration and monitoring has been done. At the moment, it is not clear how much of the information in this paragraph is from the Rossi and Lozano paper and how much is your interpretation.

The interpretations are based on Rossi and Lozano (2016) and Rossi et al. (2018). This has been clarified in the manuscript.

Section 3.1 – Material: Was the mineralogy of the sample measured? Which method was used? Based on the measurement, what is the mineralogy of the sample?

The SIR-14 stalagmite is composed entirely of calcite. The texture is identical to that of other stalagmites from the same chamber (including SIR-1: Rossi et al., 2018), some of which have been studied under thin section by Carlos Rossi, and all consist of calcite. The measured Mg/Ca ratios are too high to be aragonite: Mg is easily rejected from the orthorhombic lattice of aragonite, unlike the calcite lattice. In this case, we can use the Mg/Ca ratio as an indicator of the mineralogy.

Sections 3.3 and 3.4: Were stable isotopes and trace elements measured at the same resolution. Figure 4 suggests somewhat lower resolution trace element measurements.

No, the stable isotopes and the trace elements do not have identical resolution. The stable isotopes are sampled continuously at every 1 mm. The trace elements were measured from residual powders from the stable isotope samples, but we did not measure trace elements on all stable isotope samples due to cost limitations. We selected the trace element samples according to the aims of the paper, i.e. from ~8.3 to ~7.3 ka we measured trace elements on every stable isotope sample (i.e. each mm), whilst during the rest of the studied period the resolution varied from every $2^{nd}$ mm up to every $6^{th}$ mm. We have added a clarification about this to the manuscript (section 3.4).

Table 1: Please can you show this data as cross plots in the supplementary information section. It is a better way to understand the data. It would be helpful if this could be done wherever you refer to correlation coefficients.

Thanks for the suggestion, we agree it is sometimes better to visualise correlation data. The cross plots have been added to the supplementary material (suppl. fig. S3).

Line 260: Perhaps the 30-point running correlation won't be necessary if the data is examined between U-Th dates.

The 30-point running correlation (interval chosen on the basis that a sample size of n=30 is sufficient to draw statistical inferences about the sample population) shows the long-term trends that are not obvious in any point-to-point correlation. Since we strongly suggest the growth rate data are a robust metric (see arguments above), we retain the 30-point correlation coefficient data.

Figure 4: This is a tricky one. The anomaly in magnitude and duration is ever so small in all the proxies apart from the growth rate. The growth rate is the one that is expected to show an anomaly. I would be curious to see how the results would be after measuring growth rate between uranium-thorium dates. I also find it more intuitive to see warmer and wetter directed up. Perhaps the proxy plotting direction is this way to accommodate for the interpretation of the d18O isotopes. Maybe you could add 'interpretative keys' to the sides of the records e.g. d18O = meltwater i.e. source water change / longer travel / upstream rainout etc. or something of the kind.

The reviewer is correct. As we also point out in the manuscript, the magnitude of the anomaly is very small in most of the proxies. We have chosen to display the proxies this way as it underlines the following points: (1) highlighting the proxies that covary (or do not covary); (2) to clarify that the proxies are controlled by the same factor, i.e. effective recharge, with more recharge leading to lower d18O, d13C and Mg/Ca values; and (3) to show that d18O is leading

the other proxies during the event, emphasising that the d18O is likely to additionally be driven by a change in the isotopic composition of the moisture source.

Figure 5 description: I am wary of language like 'well-defined excursion'. The plot only covers the short duration from 9.2 to 7.8 ky. Even within the plot, neither the magnitude nor the duration of the d18O excursion stand out very clearly against the rest of the record.

We have modified the description.

Line 445: Reference Fig. S5 here. What are the pale lines in Figure S5?

Line 445 is in the figure 5 caption in the original manuscript. In figure 5 we show the SIR-14 proxies. Figure S5 is a comparison between the d18O and d13C of SIR-14 (this study) and SIR-1 (Rossi et al., 2018), and it is unclear why this figure should be referenced here. The pale lines in figure S5 (now figure S6) are the 'slow growth' periods in SIR-14 (which has been interpreted to not be related to regional climate changes). We have now clarified this in the figure caption.

Line 505: 'prominent multi-pronged decrease' again I would be wary of using such strong language.

We have modified the description.

Line 510: It could be change at moisture source and/or an increase in the distance of the moisture source from the cave location given circulations changes that maybe expected with such an event.

We have added a sentence at the end of the same paragraph raising this possibility.

Line 515: Perhaps the Stoll et al paper (https://doi.org/10.1038/s41467-022-31619-3) is useful for thinking through how the location of meltwater release may impact the oxygen isotopic composition of the source region.

Stoll et al. discuss glacial Termination II and infer the importance of water sourced from the Eurasian Ice Sheet. Whilst this is important work, we do not believe it is directly relevant to the present study, which relates to a much smaller meltwater pulse during the Holocene that was sourced from the Laurentide ice sheet (North America).

Line 520: Is the mechanism of …more effective recharge = low d13C = low Mg/Ca = low growth rate… does it not apply for slow growth phases of the speleothem? What is the 'exception' here?

More effective recharge leads to low d13C, Mg/Ca and high growth rate, and *vice versa*. These 'slow growth phases' of the speleothem have been described and explained in section 5.1.1. During these intervals, the growth is much slower and the d18O, d13C and Mg/Ca much higher compared to the rest of the record. This convergence of all four proxies suggests a hydrologically driven growth-rate reduction. During one of these events, growth in the SIR-1 stalagmite stopped altogether (at ~7.7 ka BP: Rossi et al., 2018), supporting the notion that slower growth in SIR-14 at this time could have been caused by a shift to drier conditions. However, no such evidence is found in SIR-1 to support the same interpretation for the older

slow-growth interval in SIR-14 (suppl. fig. S5). This suggests that non-climatic effects specific to the drip site may have been responsible (which would give climate-like responses to these proxies). The absence of contemporaneous climate changes in other regional palaeorecords during both of SIR-14's slow-growth periods implies that a localised perturbation of infiltration may have affected the two (or more) closely situated drip points (fig. 2). Thus, the climatic significance of the two slow-growth phases remains equivocal.

Line 545: Add acronym for LAO here if you are going to use it later in the section.

Thanks, the text has been changed using the acronym.

Data availability: Maybe best to submit data to NOAA – more findable. And Zenodo or a University repository in SISAL format with the additional metadata since SISAL database updates are not frequent.

This is a good point, we will make our data available at a more recently updated database, such as NOAA.

---

## Author Comment (AC2)

**Review "The 8.2 ka event in northern Spain: timing, structure and climatic impact from a multi-proxy speleothem record" by Kilhavn et al.**

This manuscript presents a speleothem record from northern Spain (El Soplao cave) that covers the 8.2 ka event with a well-stablished chronology. The record was presented in a previous paper focused on other time period (Rossi et al., 2018) and the chronology has now been improved. The main highlight is the combination of proxies to really infer the climate signal in this region as response to the 8.2 ka, combining growth rate, stable isotopes and trace elements with adequate resolution. There is a nice discussion to interpret the proxies and an excellent comparison with other speleothem records from W Mediterranean. The authors conclude that this event was synchronous in Greenland and S European records, with similar structure. I just have few comments that can be easily solved in a new version of the manuscript, previously to its acceptance.

We thank reviewer 2 for very positive and constructive comments and suggestions. We just wish to clarify that this record (SIR-14) has not been presented in the previous paper of Rossi et al. (2018), which focused on another stalagmite (SIR-1) located in the same cave chamber. However, several new U-Th dates were measured on SIR-1 to improve its chronology for a better comparison of its Holocene growth phase with the SIR-14 record.

1. The influence of temperature and amount of precipitation in the rainfall isotopic composition (i.e. $d^{18}O$) is not easy to determine in this region. I like the approach of separating both influences as it is presented in Fig. S2 (lines 112-115 in the main text) but I think that, from the graphs, an acceptable correlation with temperature can be inferred, excepting for samples with very high precipitation. I think those samples may correspond to heavy summer storms that can provide very negative values although temperature is high. The authors may want to check it. Therefore, this figure and the obtained correlations need a more detailed consideration and probably giving a more important role to temperature variation.

The reviewer makes a good point. There are stronger correlations (statistically significant) for temperature vs d18O for the lower bands of rainfall (less than 100 mm amount of monthly rainfall). From the GNIP data the maximum effect of temperature on d18O variation is 28 % ($r = 0.53$). Thus, it appears that the temperature effect is potentially important when there is less precipitation, however, it cannot alone explain the variations in d18O. We have added this to our discussion about driving factors for the d18O (see section 5.1.4).

2. Besides, in line 465, it is considered a $d^{18}O$ – temperature gradient between 0.24 – 0.34 ‰/°C, following GNIP results presented in (Domínguez-Villar et al., 2008), values that can be higher in other areas in northern Spain (please, check Moreno et al., 2021 for information at event-scale). If those values are higher, they won't be counterbalanced by the temperature dependence of water-calcite isotope fractionation in the cave. Thus, I would not exclude so rapidly temperature as an important influence on $d^{18}O$ record. I think that temperature influence can be higher than 0.11 ‰/°C as pointed the authors in line 468. Still, I agree with the authors that very likely, the effective recharge was a more important factor on $d^{18}O$ values.

We thank reviewer 2 for pointing us to this paper: Moreno et al. (2021). However, Moreno et al. suggest a temperature effect of maximum +0.38 ‰ °C$^{-1}$, which is not significantly higher than the estimate from Domínguez-Villar et al. (2008), and this would still be partially

counterbalanced by the temperature dependence of water-calcite isotope fractionation in the cave (leaving a residual of +0.14 ‰ °C$^{-1}$). The magnitude of variability in SIR-14 d18O (from -5.9 to -4.3 ‰), and the known low temperature variability of the Holocene, would relegate temperature to being of minor influence. Nevertheless, we have added this paper and the additional information about temperature gradients from other sites in Spain to the manuscript (section 5.1.4).

3. Although I agree that other records such as lake or marine sediments lack the adequate resolution (in the sampling and in the chronology) to provide information about the timing of the 8.2 ka, I don't agree about neglecting the information they can offer on the impact of that event. I think that information can be of importance to get the regional picture and try to stablish the forcing mechanisms. It is important to include some lacustrine records and archaeological sites in the discussion section 5.2.3 since they are indicating, in general, a dry period during the 8.2 ka event, contrarily to what is observed in the speleothem records. I would recommend checking the Basa de la Mora record (a well-dated Holocene record from a lake in the Central Pyrenees) (Pérez-Sanz et al., 2013); the pollen record from marine core MD952043 and references therein (Fletcher et al., 2013) and a compilation of archaeological sites from the Ebro valley that were abandoned during the 8.2 ka due to dry conditions (González-Sampériz et al., 2009). There is also a recent paper on this topic (García-Escárzaya et al., 2022). I think all these records will enrich the discussion and may allow to define different regions in Iberia with distinct responses to the 8.2 ka event.

We thank reviewer 2 for this suggestion. However, as the reviewer points out, most of these other records lack the adequate resolution to provide information about the timing of the event. Additionally, most of these other records show a much longer-lasting climate anomaly, typically spanning ~300-400 years. Thus, it is likely that (at least) some of these records are not showing a response to the short-lived 8.2 ka climate event but are rather linked to summer insolation (as pointed out by Morellón et al., 2018). There seems to be an overall consensus towards drier conditions in regions associated with the Mediterranean and more humid conditions in regions associated with the North Atlantic in the early Holocene (although there are exceptions), suggesting that there is a different response in different regions. However, without more precisely dated records that capture this difference in climatic response lasting for ~150 years (the average duration of the 8.2 ka event), it would make things ambiguous to include these records. Nevertheless, we have added some additional discussion regarding other records in the region, as we agree with the reviewer, they offer important information about the climate in the early Holocene.

Minor comments:

- I miss the age model figure for SIR-1

The age-model for SIR-1 is shown in the supplementary material, figure S8D.

- I am surprised that generating a new chronology for the presented records provides such differences in timing comparing with the previous ones (more than 200 years of temporal shift in some cases). This is important to me since considering one or the other way of generating the age model makes the 8.2 ka event to be synchronous or not. I wonder if the authors considered to improve the chronologies with more dates, not only with a different modelling approach to get a more robust approach here.

The reviewer is right in pointing this out: the new age models can make the 8.2 ka event to be synchronous or not. However, we created the new chronologies independently of their associated proxy records to avoid potential biases in tuning the records to one another. The new chronologies were created by using the same approach described by Corrick et al. (2020). First, the U-Th ages were recalculated using the most recent estimates of the decay constants (Cheng et al., 2013) and by modelling the initial 230Th/232Th activity. These calculations were conducted by co-author John Hellstrom, who was not privy to the proxy data. Lastly, these recalculated ages were used to create the new chronology by the Finite Positive Growth Rate Model. To improve the chronologies with more dates would be the ideal way to go to test the synchrony of the event. However, it is beyond the scope of this paper to refine the dating of the selected published records. These were already selected based on several criteria and considered of high-quality (i.e., sufficient resolution and well-constrained chronologies).

- Line 691: the reference Zielhofer et al., 2019 does not correspond to SW Europe (it may be better to talk about W Mediterranean).

We have modified the text here.

References

Domínguez-Villar, D., Wang, X., Cheng, H., Martín-Chivelet, J., and Edwards, R. L.: A high-resolution late Holocene speleothem record from Kaite Cave, northern Spain: d18O variability and possible causes, Quaternary International, 187, 40–51, 2008.

Fletcher, W. J., Debret, M., and Goñi, M. F. S.: Mid-Holocene emergence of a low-frequency millennial oscillation in western Mediterranean climate: Implications for past dynamics of the North Atlantic atmospheric westerlies, The Holocene, 23, 153–166, https://doi.org/10.1177/0959683612460783, 2013.

Asier García-Escárzaga, Igor Gutiérrez-Zugasti, Ana B. Marín-Arroyo, Ricardo Fernandes, Sara Núñez de la Fuente, David Cuenca-Solana, Eneko Iriarte, Carlos Simões, Javier Martín-Chivelet, Manuel R. González-Morales & Patrick Roberts Human forager response to abrupt climate change at 8.2 ka on the Atlantic coast of Europe, Scientific Reports volume 12, 6481, 2022.

González-Sampériz, P., Utrilla, P., Mazo, C., Valero-Garcés, B., Sopena, M. C., Morellón, M., Sebastián, M., Moreno, A., and Martínez-Bea, M.: Patterns of human occupation during the Early Holocene in the Central Ebro Basin (NE Spain) in response to the 8.2 ka climatic event, Quaternary Research, 71, 121–132, 2009.

Moreno, A., Iglesias, M., Azorin-Molina, C., Pérez-Mejías, C., Bartolomé, M., Sancho, C., Stoll, H., Cacho, I., Frigola, J., Osácar, C., Muñoz, A., Delgado-Huertas, A., Bladé, I., and Vimeux, F.: Measurement report: Spatial variability of northern Iberian rainfall stable isotope values – investigating atmospheric controls on daily and monthly timescales, Atmospheric Chemistry and Physics, 21, 10159–10177, https://doi.org/10.5194/acp-21-10159-2021, 2021.

Pérez-Sanz, A., González-Sampériz, P., Moreno, A., Valero-Garcés, B., Gil-Romera, G., Rieradevall, M., Tarrats, P., Lasheras-Álvarez, L., Morellón, M., Belmonte, A., Sancho, C.,

Sevilla-Callejo, M., and Navas, A.: Holocene climate variability, vegetation dynamics and fire regime in the central Pyrenees: the Basa de la Mora sequence (NE Spain), Quaternary Science Reviews, 73, 149–169, https://doi.org/10.1016/j.quascirev.2013.05.010, 2013.

Rossi, C., Bajo, P., Lozano, R. and Hellstrom, J. Younger Dryas to Early Holocene paleoclimate in Cantabria (N Spain): Constraints from speleothem Mg, annual fluorescence banding and stable isotope records, Quaternary Science Reviews, 192, 71-85 2018 https://doi.org/10.1016/j.quascirev.2018.05.025.